# Modelling organic aerosol concentrations and properties during ChArMEx summer campaigns of 2012 and 2013 in the western Mediterranean region

Mounir Chrit[1], Karine Sartelet[1], Jean  Sciare[2,7], Jorge Pey[3*], Nicolas Marchand [3], Florian Couvidat[4], Karine Sellegri[5], and Matthias Beekmann[6]

[1]CEREA, joint laboratory Ecole des Ponts ParisTech - EDF R&D, Université Paris-Est, 77455 Champs sur Marne, France.

[2] LSCE, CNRS-CEA-UVSQ, IPSL, Univ. Paris-Saclay, Gif-sur-Yvette, France

[3] Aix Marseille University, CNRS, LCE UMR 7376, Marseille, France

[4] INERIS, Verneuil en Halatte, France

[5] LAMP, Aubière, France

[6] LISA, UMR CNRS 7583, IPSL, Université Paris Est Créteil Créteil and Université Paris Diderot, France

[7] EEWRC, The Cyprus Institute, Nicosia, Cyprus

[*] Now at the Geological Survey of Spain, IGME, 50006 Zaragoza, Spain

*Correspondence to:* Mounir Chrit (mounir.chrit@enpc.fr)

**Abstract.**

In the framework of the Chemistry-Aerosol Mediterranean Experiment, a measurement site was set up at a remote site (Ersa) on Corsica Island in the northwestern Mediterranean Sea. Measurement campaigns performed during the summers of 2012 and 2013 showed high organic aerosol concentrations, mostly from biogenic origin. This work aims at representing the organic aerosol concentrations and properties (oxidation state and hydrophilicity) using the air-quality model Polyphemus with a surrogate approach for secondary organic aerosol (SOA) formation. Biogenic precursors are isoprene, monoterpenes and sesquiterpenes. In this work, the following model oxidation products of monoterpenes are added: (i) a carboxylic acid (MBTCA) to represent multi-generation oxidation products in the low-NOx regime, (ii) organic nitrate chemistry, (iii) extremely low volatility organic compounds (ELVOCs) formed by ozonolysis. The model shows good agreement to measurements of organic concentrations for both 2012 and 2013 summer campaigns. The modeled oxidation property and hydrophilic organic carbon properties of the organic aerosols also agree reasonably well with the measurements. The influence of the different chemical processes added to the model on the oxidation level of organics is studied. Measured and simulated water-soluble organic concentrations (WSOC) show that even at a remote site next to the sea, about 64% of the organic carbon is soluble. The concentrations of WSOC vary with the origins of the air masses and the composition of organic aerosols. The marine organic emissions only contribute to a few percent of the organic mass in $PM_1$, with maxima above the sea.

## 1  Introduction

The Mediterranean region is considered as one of the prominent regions that could be detrimentally impacted by climate and air composition changes over both southern Europe and northern Africa. Organic aerosols (OA) account for about 20-50% of

the fine aerosol mass at continental mid latitudes (Saxena and Hildemann, 1996) and as high as 90% in tropical forest areas (Kanakidou et al., 2005; Jimenez et al., 2009). They contribute to more than 50% of EU-regulated $PM_{2.5}$ concentrations in Europe (Putaud et al., 2004). OA affect both climate and human health. They influence the radiation budget by mostly scattering sunlight resulting in negative direct radiative forcing (Fuzzi et al., 2006; Lin et al., 2014). Moreover, hydrophilic OA can act as a cloud condensation nuclei and hence modify cloud microphysical properties and lifetime. In terms of health effects, OA toxicity is linked to the oxidative stress which is induced by the ROS (Reactive Oxygen Species). The oxidative potential may differ for the different organic precursors (Rattanavaraha et al., 2011; Jiang et al., 2016; Tuet et al., 2017).

OA are usually classified either as primary (POA) or as secondary aerosols (SOA). POA are directly emitted in the atmosphere, often as intermediate/semi-volatile organic compounds (I/S-VOCs), which partition between the gas and the particle phases (Robinson et al., 2007). The gas-phase of I/S-VOC is missing from emission inventories (Couvidat et al., 2012; Kim et al., 2016). SOA are produced through chemical oxidation of volatile organic compounds (VOCs) and I/S-VOCs, and condensation of I/S-VOCs.

A large fraction of emitted VOCs is biogenic, especially in the western Mediterranean in summer, when solar radiation is high. Biogenic emissions may age and form SOA as they are transported through different environments (Hayes et al., 2015). Using aerosol mass spectrometer (AMS) measurements performed in an urban area in South France (Marseille) and positive matrix factorisation (PMF) techniques, El Haddad et al. (2011, 2013) attributed 80% of the organic aerosol mass to biogenic secondary organic aerosols (BSOA), and they attributed near 40% of the BSOA to monoterpene oxidation products. These high biogenic concentrations in an urban area may be partly explained by the influence of anthropogenic emissions on biogenic SOA formation (Carlton et al., 2010; Hoyle et al., 2011; Sartelet et al., 2012).

Similar results were obtained through measurement campaigns in the Barcelona region (Spain), where Minguillón et al. (2011, 2016) have found a prevalence of non-fossil organic aerosol sources in remote and urban environments, and a clear evidence of biogenic VOC oxidation products and biogenic SOA formation under anthropogenic stressors.

The ChArMEx (Chemistry-Aerosol Mediterranean Experiment: http:// charmex.lsce.ipsl.fr) project has organized several summer campaigns to study atmospheric chemistry and its impacts in the western Mediterranean region. The TRAQA (Transport et Qualité de l'Air) campaign was set up in summer 2012 to study the transport and impact of continental air on atmospheric pollution over the basin (Sic et al., 2016). The ADRIMED (Aerosol Direct Radiative Impact in the Mediterranean; Mallet et al. (2016)) campaign was set up in June-July 2013 to assess the radiative impact of aerosols, while the SAF-MED (Secondary Aerosol Formation in the MEDiterranean) campaign was set up July-early August 2013 to understand and characterize the concentrations and properties of organic aerosols in the western Mediterranean and to figure out the origins of the high concentrations observed outside urban areas. Intensive ground-based in-situ measurements were performed during the summer 2013, while airborne measurements were performed during the summers 2013 and 2014 (Di Biagio et al., 2015; Freney et al., 2017). In agreement with the observations of El Haddad et al. (2013) in Marseille on the French Mediterranean coast, the ground-based in-situ measurements performed at the remote site of Ersa on Cape Corsica, the northern tip of Corsica Island (South-East of continental France) showed that OA are mostly from biogenic origin. However, as over urban areas,

anthropogenic emissions, from shipping or pollution plumes from European big cities, may influence biogenic SOA formation (Sartelet et al., 2012).

The VOC biogenic precursors of SOA are isoprene, monoterpenes and sesquiterpenes. Although sesquiterpenes emission factors are lower than those of isoprene and monoterpenes over Europe, their SOA yields are high because of their low saturation vapour pressures (Jaoui et al., 2013). For monoterpenes, first-generation oxidation products, such as pinonaldehyde, pinic and pinonic acids contribute to the formation of SOA, although their contributions may be low (Praplan et al., 2015). For $\alpha$-pinene, further-generation oxidation steps may lead to the formation of very low volatile products, such as the tricarboxylic acid 3-methyl-1,2,3-butanetricarboxylic acid (MBTCA, Müller et al. (2012); Kristensen et al. (2014)) and oligomeric compounds. Ehn et al. (2014) and Kristensen et al. (2014) showed that highly-oxidised organic compounds are formed in the early stages of the oxidation of monoterpenes. Rissanen et al. (2015) proposed a mechanistic description of these extremely low volatile organic compounds (ELVOCs) formation from the most atmospherically abundant biogenic monoterpenes, such as $\alpha$-pinene and limonene. These ELVOCs have been observed both during chamber and in-situ measurements in Germany (Mutzel et al., 2015). Several studies showed the importance of nighttime SOA formation from monoterpenes via nitrate radical oxidation, resulting in the formation of organic nitrates (Pye et al., 2015; Bean and Ruiz, 2016; Xu et al., 2015; Nah et al., 2016). Pye et al. (2010) showed the importance of reactive nitrogen and pointed out the fact that organic nitrate accounts for more than a half of the monoterpene oxidation products in the particle phase over the U.S..

For isoprene, in low-NOx environments, recent studies have focused on the formation of isoprene epoxydiols (IEPOX) in acidic aerosols (Surratt et al., 2010; Couvidat et al., 2013b). However, using aerosol mass spectrometer measurements, particle-phase IEPOX was not observed during the ChArMEx campaign over an isoprene-emitting forest in the South of France, suggesting it might have formed organo-sulfates (Freney et al., 2017). Several studies also showed the importance of non-IEPOX pathway for isoprene oxidation in low-NOx environment (Krechmer et al., 2015; Liu et al., 2016). Although ELVOCs may also form from isoprene oxidation, the yields may be low (Jokinen et al., 2015).

OA can also be emitted from the sea, because of phytoplankton activity: according to O'Dowd et al. (2004), OA from marine origin can contribute considerably to OA concentrations especially near the biologically productive waters. Recently derived parameterisations (Gantt et al., 2012; Schwier et al., 2015) relate the organic fraction of sea-salt emissions to the seawater chlorophyll-a concentrations. However, the contribution of these species to the organic budget over the Mediterranean Sea is not clear.

SOA modeling has undergone significant progress over the past few years due to the rapid increase of experimental data on SOA yields and molecular chemical composition resulting from the oxidation of a variety of VOC and I/S-VOC. SOA models used in meso-scale models can be grouped into two major categories: (1) models based on an empirical representation of SOA formation and (2) models based on a mechanistic representation of SOA formation. Models of the first category include the widely used two-compound Odum approach (Odum et al., 1996) and the more recent volatility basis set (VBS) approach (Donahue et al., 2006, 2011), or the multi-generational oxidation model (Jathar et al., 2015). Models of the second category use experimental data on the molecular composition of SOA and represent the formation of SOA using surrogate molecules with representative physico-chemical properties (Pun et al., 2006; Bessagnet et al., 2008; Carlton et al., 2010; Couvidat et al., 2012).

The surrogate approach differentiates low-NOx and high-NOx regimes. The gas/particle partitioning may include both absorption into hydrophobic organic particles and dissolution into aqueous particles, and take into account some of the complexity involved in OA partitioning (such as non-ideality, multi-phase partitioning). Although these two categories of models are fundamentally different in their initial design (empirical vs. mechanistic), they aim at describing the same processes. Furthermore, they tend to converge as they continue to be developed and refined. For example, I/S-VOCs from anthropogenic emissions, which are usually specified by volatility classes, can be included in a mechanistic model (Albriet et al., 2010; Couvidat et al., 2013a), the VBS scheme can take into account the oxidative state of SOA (Jimenez et al., 2009), in particular its elemental O:C ratio (Donahue et al., 2011; Jathar et al., 2015). The recently developed 1.5-VBS (Koo et al., 2014) assigned a molecular structure to VBS products. Both the mechanistic and empirical approaches are scientifically valid and complementary; as shown by Kim et al. (2011a), the most important aspect of an SOA model is its comprehensiveness in terms of the precursors and processes being treated (completeness of the precursor VOC list, importance of low-NOx vs. high-NOx regimes, treatment of hydrophilic properties of the surrogates) rather than its fundamental design. Cholakian et al. (2017) study how the VBS approach can be used to represent the formation of SOA over the western Mediterranean and point out the importance of taking into account fragmentation and formation of non-volatile SOA in this framework.

This paper aims at investigating the chemical processes and surrogates that need to be taken into account in the mechanistic representation of SOA to reproduce the concentrations and properties of the observed biogenic SOA at the Ersa super-site in Corsica. The mechanistic representation included in the air-quality model Polyphemus (Couvidat et al., 2012) is modified by including recent research progress on monoterpenes SOA formation (ELVOC, MBTCA, organic nitrate). The influence of primary marine organic emissions is also studied. A further evaluation of the model by comparison to airborne measurements is presented in Freney et al. (2017).

The paper is structured as follows. Section 2 presents the air-quality model used as well as the improvements made in the mechanistic representation. Section 3 details the model input data-sets and the measurement data. Section 4 compares the concentrations and the properties of OA to measurements, and the influence of the different chemical processes added to the model. Finally, section 5 studies the impact of the biological activity of the Mediterranean Sea on OA concentrations.

## 2   Model description

### 2.1   General features

In order to simulate aerosol formation over the western Mediterranean, the Polair3d/Polyphemus air-quality model is used (Mallet and Sportisse, 2005). The numerical algorithms used for transport, and the parameterisations used for dry and wet depositions are detailed in Sartelet et al. (2007). Gas-phase chemistry is modeled with the Carbon Bond 05 mechanism (CB05) (Yarwood et al., 2005). Different reactions are added to CB05 to model the formation of semi-volatile organic compounds from five classes of SOA precursors (intermediate and semi-volatile organic compounds of anthropogenic emissions, aromatic compounds, isoprene, monoterpenes and sesquiterpenes) (Kim et al., 2011b; Couvidat et al., 2012). For these classes of precursors, which include a great number of species, only a few surrogates are used to represent all the species.

| Precursors | Surrogate species |
|---|---|
| I/S-VOCs | 3 volatility bins: $\log(C^*)$ = -0.04, 1.93, 3.5 with $C^*$ the saturation concentration |
| Aromatics | Toluene, xylene |
| Isoprene | Isoprene |
| Monoterpenes | $\alpha$-pinene, $\beta$-pinene, limonene |
| Sesquiterpenes | Humulene |

**Table 1.** Precursors classes and the surrogate species used for SOA formation.

As detailed in Couvidat and Seigneur (2011), isoprene may tetrols and methyl dihydroxy dihydroperoxide under low NOx, and methyl glyceric acids and organic nitrates under high NOx. Oxidation of isoprene by the nitrate radical $NO_3$ is also modeled.

For monoterpenes and sesquiterpenes, the oxidation scheme is based on Pun et al. (2006). Humulene is used to represent all sesquiterpenes. For monoterpenes, three precursors are used: API (for $\alpha$-pinene and sabinene), BPI (for $\beta$-pinene and $\delta^3$-carene) and LIM (for limonene and other monoterpenes and terpenoids). Depending on the NOx regimes, three surrogates are formed: pinonaldehyde, norpinic acid and pinic acid. Although a simple parameterisation was developed to represent the oligomerization of pinonaldehyde as a function of pH in Couvidat et al. (2012), it is not used here because its influence on SOA formation is not clear. As detailed in Couvidat et al. (2012), I/S-VOC emissions are emitted as three primary surrogates of different volatilities (characterized by their saturation concentrations $C^*$: $\log(C^*)$ = -0.04, 1.93, 3.5). The ageing of each primary surrogate is represented through a single oxidation step, without $NO_x$-dependance, to produce a secondary surrogate of lower volatility ($\log(C^*)$ = -2.4, -0.064, 1.5 respectively) but higher molecular weight. For aromatic compounds, toluene and xylene are used as surrogate precursors. The precursors react with OH to form radicals that may then react differently under low-$NO_x$ and high-$NO_x$ conditions. Under low-$NO_x$ conditions, the surrogate is not identified, but it is supposed to be hydrophobic. Under high-$NO_x$ conditions, the surrogate formed are two benzoic acids (methyl nitro benzoic acid and methyl hydroxy benzoic acid). Table 1 describes the five classes of precursors used to represent the SOA formation and the surrogates used.

The SIze REsolved Aerosol Model (SIREAM) (Debry et al., 2007) is used for simulating the dynamics of the aerosol size distribution by coagulation and condensation/evaporation. SIREAM uses a sectional approach and the aerosol distribution is described here using 20 sections of bound diameters: 0.01, 0.0141, 0.0199, 0.0281, 0.0398, 0.0562, 0.0794, 0.1121, 0.1585, 0.2512, 0.3981, 0.6310, 1.0, 1.2589, 1.5849, 1.9953, 2.5119, 3.5481, 5.0119, 7.0795 and 10.0 $\mu$m. The condensation/evaporation of inorganic aerosols is determined using the thermodynamic model ISORROPIA (Nenes et al., 1998) with a bulk equilibrium approach in order to compute partitioning between the gaseous and condensed phases of particles.

For organic aerosols, the gas-particle partitioning of the surrogates is computed using SOAP (Couvidat and Sartelet, 2015), and bulk equilibrium is also assumed for SOA partitioning. The gas-particle partitioning of hydrophobic surrogates is modelled following Pankow (1994), with absorption by the organic phase (hydrophobic surrogates). The pas-particle partitioning of

| VOC | Ehn et al. (2014) | Jokinen et al. (2015) |
|---|---|---|
| $\alpha$-pinene | 7% $\pm$ 3.5% | 3.4% $\pm$ 1.7% |
| limonene | 17% $\pm$ 8.5% | 5.3% $\pm$ 2.6% |

**Table 2.** ELVOC yields and uncertainties.

hydrophilic surrogates is computed using the Henry's law modified to extrapolate infinite dilution conditions to all conditions using an aqueous-phase partitioning coefficient, with absorption by the aqueous phase (hydrophilic organics, inorganics and water). Activity coefficients are computed with the thermodynamic model UNIFAC (UNIversal Functional group, Fredenslund et al. (1975)). After condensation/evaporation, the moving diameter algorithm is used for mass redistribution among size bins.

## 2.2 ELVOCs

Ehn et al. (2014) and Kristensen et al. (2014) showed that hydrophobic high-molecular weight molecules of extremely low volatility form at the early stage of oxidation of the monoterpenes $\alpha$-pinene and limonene by ozone. In the model, a unique gaseous precursor representing $\alpha$-pinene and limonene is used for ELVOC formation. The ELVOC yield is assumed to be 11%, i.e. close to the average of the yields of $\alpha$-pinene and limonene according to Ehn et al. (2014). Jokinen et al. (2015) suggested lower yields (Table 2). In this paper, sensitivity simulations with a lower bound of 3% and a upper bound of 18% are also conducted.

Relying on known chemistry and experimental findings, Ehn et al. (2014) and Rissanen et al. (2015) provided a formation pathway from monoterpenes to ELVOCs through the autoxidation process (Crounse et al., 2013).

The ozonolysis reaction leads to the formation of peroxy radicals ($RO_2$), which are the starting point of ELVOCs formation. These radicals undergo a fairly rapid ($\sim 1~s^{-1}$) sequential intramolecular H-atom shift followed by $O_2$ addition leading to the formation of highly-oxygenated peroxy radicals ($R_{ELVOC}O_2$). This $O_2$ addition leads not only to an increase in molecular weight, but also to a decrease in the radical volatilities. Rissanen et al. (2015) investigated these reactions including the different steps, possible isomerizations as well as the most likely chemical pathways leading to an enrichment of peroxy radicals by oxygen. The oxygen-centered peroxy radical intermediates are internally rearranged by intramolecular hydrogen shift reactions, enabling more oxygen molecules to attach to the carbon backbone. Simultaneously, the sequential H-shift mechanism competes with reactions between peroxy radicals, NO and $HO_2$. Subsequently, the termination reactions lead to the formation of ELVOCs. ELVOC monomers and dimers are formed through reactions of $R_{ELVOC}O_2$ with $RO_2$ and $HO_2$. Following Ehn et al. (2014), the reactions for the ozonolysis of $\alpha$-pinene and limonene (monoterpenes, MT) are included in the model as detailed in Appendix A, as well as the kinetic constants used in the model (Table[A1] in Appendix A). The formation of organic nitrate from $R_{ELVOC}O_2$ peroxy radicals is not considered in the model, as the reactions proposed by Ehn et al. (2014) led to negligible concentrations.

| Species | Molecular formula | Molar weight [g.mol$^{-1}$] | Saturation vapour pressure at 298 K [torr] | Enthalpy of vapourisation [KJ.mol$^{-1}$] | OM/OC | O/C |
|---|---|---|---|---|---|---|
| Monomer ELVOC | $C_{10}H_{14}O_9$ | 278 | $1.0\ 10^{-14}$ | 50.0 | 2.3 | 1.2 |
| Dimer ELVOC | $C_{19}H_{28}O_{11}$ | 432 | $1.0\ 10^{-14}$ | 50.0 | 1.9 | 0.8 |
| org$_{NIT}$ | $C_{10}H_{17}NO_5$ | 231 | $5.0\ 10^{-6}$ | 40.0 | 1.9 | 0.7 |
| MBTCA | $C_8H_{12}O_6$ | 204 | $3.25\ 10^{-7}$ | 109 | 2.1 | 1.0 |
| SSorg | | 136 | $6.60\ 10^{-8}$ | 50.0 | | |

**Table 3.** Species introduced in the ELVOC, organic nitrate and MBTCA kinetic models.

The aerosol species introduced in the model for ELVOC formation and their properties are detailed in Table [3]. The enthalpy of vapourisation of the monomer and dimer is set to 50 KJ.mol$^{-1}$ (Svendby et al., 2008), and the saturation vapour pressure is assumed to be very low and is taken equal to $10^{-14}$ torr at 298 K.

### 2.3 Organic nitrates formation mechanism

Organic nitrates are formed where biogenic VOCs and anthropogenic NO$_x$ sources interact (Pye et al., 2015; Xu et al., 2015; Bean and Ruiz, 2016; Nah et al., 2016). We used here the parameterisation of Pye et al. (2015) to account for the formation of organic nitrate compounds from the oxidation by OH and NO$_3$ of monoterpenes (MT). The oxidation of MT by OH leads to the formation of a peroxy radical TERPRO$_2$, and the oxidation by NO$_3$ leads to the formation of TERPNRO$_2$ (night-time chemistry). The peroxy radicals TERPRO$_2$ may react with NO to form organic nitrate with a molar yield of 20.1%, while the

oxidation of TERPNRO$_2$ leads to higher yields. The reactions are described in Appendix B, as well as the kinetic constants (Table [B1] in Appendix[B]).

Following Pye et al. (2015), the estimated vapour pressure of the condensing organic nitrate species is assumed to be $5.10^{-6}$ torr (Fry et al., 2009), and the enthalpy of vapourisation is taken as 40 KJ.mol$^{-1}$. The aerosol species (orgNIT) introduced in the model and its properties are summarized in Table[3]. The organic nitrate is assumed to be hydrophobic (Liu et al., 2012).

### 2.4 MBTCA: an aging product of the pinonic acid

It was shown in a set of studies that ozonolysis and OH-initiated reactions of terpenes produce organic acids (Hatakeyama et al., 1991; Hoffmann et al., 1997; Warnke et al., 2006). Szmigielski et al. (2007) identified MBTCA (3-methyl-1,2,3-butanetricarboxylic acid) as the most relevant organic acid for atmospheric SOA. It is produced by the OH-oxidation of pinonic acid, which is itself produced by the OH-oxidation of $\alpha$-pinene. The OH-oxidation of pinonic acid to form MBTCA is added

to the model with a kinetic constant k = $9.0 \times 10^{-12}$ cm$^3$.s$^{-1}$ (Jaoui and Kamens, 2001) and a yield of 0.0061 (Müller et al., 2012). Following Couvidat et al. (2012), MBTCA is supposed to be hydrophilic, with a OM/OC ratio of 2.125 (Table 3).

## 3 Model and measurement setup

The simulation domain and the input data are now detailed, as well as the measurements used in this study.

### 3.1 Model setup

#### 3.1.1 Domains

Two nested simulations are performed: one over Europe and one over a Mediterranean domain centered around the Ersa super-site surroundings in Corsica (Figure 1).

The coordinates of the European southwestern-most point are (15°W , 35°N) in longitude/latitude. The domain of simulation covers an area of $50° \times 35°$ with a uniform spatial step of $0.5°$ along both longitude and latitude. For the nested Mediterranean domain, the southwestern-most point is (4°W , 39°N) in longitude/latitude. The domain of simulation covers an area of $11° \times 8°$ with a uniform spatial step of $0.125°$ ($\sim$13 km) along both longitude and latitude. 14 vertical levels are considered from the ground to 12 km. The heights of the cell interfaces are 0, 30, 60, 100, 150, 200, 300, 500, 750, 1000, 1500, 2400, 3500, 6000, 12000 m.

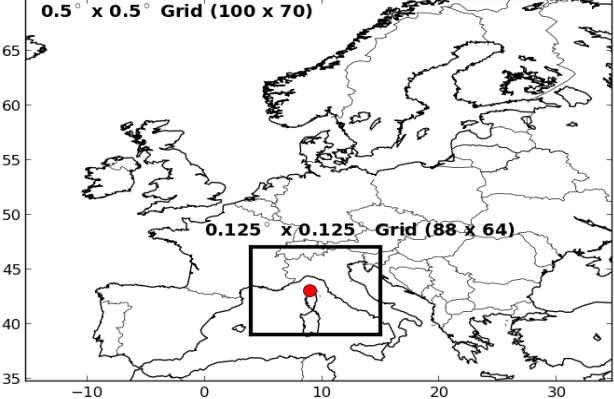

**Figure 1.** The nested modeling domains: the nesting domain over Europe and a nested domain over the northwestern Mediterranean, as delimited by a black rectangular contour in the figure. The red point indicates the Ersa station in Corsica Island.

The dates of simulation are chosen such as matching those of the measurements. For 2012, the simulations are run between 2 June and 8 July 2012 (6 June and 8 July 2012 respectively) for the nesting (nested respectively) domains. For 2013, the simulations are run between 2 June and 10 August 2013 (6 June and 10 August 2013 respectively) for the nesting (nested respectively) domains.

Boundary conditions for the European domain are obtained from the global chemistry-transport model MOZART v4 (Horowitz et al., 2003) (https://www.acom.ucar.edu/wrf-chem/mozart.shtml). The European simulation provides initial and boundary conditions to the Mediterranean simulation.

### 3.1.2 Meteorological data

Meteorological data are provided by the European Center for Medium-Range Weather Forecasts (ECMWF) model. The vertical diffusion is computed using the Troen and Mahrt parameterisation (Troen and Mahrt, 1986). The Global Land Cover 2000 (GLC-2000; http://www.gvm.jrc.it/glc2000/) data set is used for land cover.

### 3.1.3 Emissions

Anthropogenic emissions are generated using the EDGAR-HTAP_V2 inventory for 2010 (http://edgar.jrc.ec.europa.eu/htap_v2/).
The monthly and daily temporal distribution for the different activity sectors are obtained from GENEMIS (1994), and the hourly temporal distribution from Sartelet et al. (2012). Following Sartelet et al. (2007), $NO_x$ emissions are split in mass into 90% of NO, 9.2% of $NO_2$ and 0.8% of HONO. $SO_x$ emissions are split into 98% of $SO_2$ and 2% of $H_2SO_4$ (in molar concentrations). For emissions of non methane volatile organic compounds, the speciation of Passant (2002) is used. $PM_{2.5}$ primary particle emissions are speciated into dust, primary organic aerosols (POA) and black carbon (BC). POA are assumed
to be the particle phase of I/S-VOC. Total I/S-VOC emissions (gas and particle phases) are estimated as detailed in Couvidat et al. (2012), by multiplying POA by a fixed value, and by assigning them to species of different volatilities. The volatility distribution is kept the same for all emission sectors, although more detailed volatility distributions could be defined following the work of May et al. (2013a, b); Jathar et al. (2014). In this study, the ratio I/S-VOC/POA is set to 2.5 (Kim et al., 2016; Zhu et al., 2016). Setting the ratio SVOC/POA to 1, i.e. ignoring I/S-VOC, has little impact on the organic concentrations, as shown
in Figure 2. Particles of diameters higher than 2.5 $\mu$m are all speciated into dust. Biogenic emissions are estimated with the Model of Emissions of Gases and Aerosols from Nature (MEGAN, Guenther et al. (2006)). Over the Mediterranean domain, during the period of the 2013 summer simulation, the average emissions of sesquiterpenes, monoterpenes and isoprene are 0.001, 0.019 and 0.024 $\mu$g.m$^{-2}$.s$^{-1}$ respectively. Hence, comparing to isoprene and monoterpene emissions, the sesquiterpene emissions are lower by a factor of 95.8% and 94.7% respectively.

Sea-salt emissions are parameterised following Jaeglé et al. (2011), which models the generation of sea salt by the evaporation of sea spray produced by bursting bubbles during whitecap formations due to the friction with surface wind. The emitted sea-salt mass is assumed to be made of 30.61% sodium (Seinfeld and Pandis, 2006), 25.40% chloride and 4.22% sulfate following results from measurements in mesocosms made in Corsica in July 2012 (Schwier et al., 2015). The organic fraction of sea-salt emissions is not taken into account in the simulation presented here. However, it is estimated in section 5, where the
contribution of organic sea-salt emissions to organic concentrations is assessed.

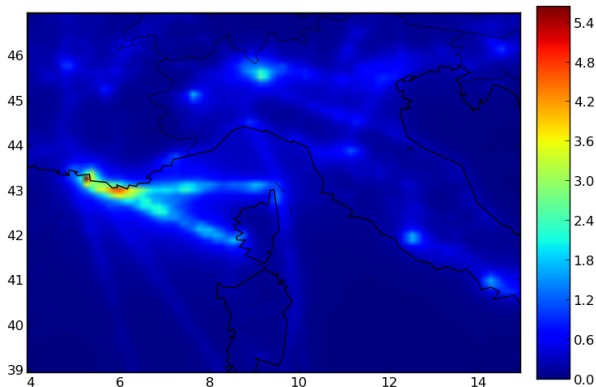

**Figure 2.** Relative difference (%) of OM$_1$ concentrations simulated using the emission ratio I/S-VOC/POA = 2.5 and 1.

## 3.2 Measured data

The model results are compared to observational data from ChArMEx campaigns during the summers of 2012 and 2013. The station is located at the red point in Figure [1]. The measurement site is located at Ersa (42°58'N, 9°21.8'E), it is located on a ridge at an altitude of about 530 m above the sea level and has an unimpeded view of the sea over ~300° from the SSW to SSE (Mallet et al., 2016). The ground-based comparisons are performed by comparing the measured concentrations to the simulated ones using the concentrations of the model cell the closest to the station. The central coordinate of this cell at which concentrations are computed is (42°52N, 9°22'30"E), which is very close to the station and with a similar altitude above sea level (494 m). Cholakian et al. (2017) studied the difficulty to correctly represent in a model the orography of Ersa site, which is a cape at the northern edge of Corsica. They concluded that the representativeness error is about 10% for organic aerosols.

To evaluate the organic concentrations and oxidation properties, an ACSM (Aerosol Chemical Speciation Monitor) was used to measure the real-time chemical composition and mass loading of aerosols with aerodynamic diameters between 70 and 1000 nm (sulfate, nitrate, ammonium, chloride and organic compounds), between 8 June and 2 July 2012, and between 6 June and 3 August 2013. The ratio OM/OC and the oxidation state of organics are estimated using the ACSM measurements following Kroll et al. (2011).

Other instruments were deployed in 2013 to evaluate the organic properties: a PILS-TOC-UV to estimate the water-soluble fraction of organics (Sciare et al., 2011) between 14 July and 5 August 2013, and a Hi-Vol quartz filter sampling DIGITEL for [14]C measurements in organics between 16 July and 30 July 2013.

A direct evaluation of the simulated concentrations of ELVOCs or organic nitrates cannot be done because they were not measured during the campaigns.

### 3.3 Model/measurements comparison method

To evaluate a model, several approaches and performance scores can be used. Here, we compare model simulation results to measurements using a set of performance statistical indicators: the simulated mean ($\bar{s}$), the root mean square error (RMSE), the correlation coefficient, the mean fractional bias (MFB), the mean fractional error (MFE). They are defined in Table [C1] of Appendix C. Based on the MFB and the MFE, Boylan and Russell (2006) proposed a performance and a goal evaluation criteria as detailed in Table [C2] of Appendix C.

## 4 Comparison to measurements

The concentrations of organic aerosols are compared to measurements for the summers 2012 and 2013. The origins of organic aerosols (fossil vs non fossil), and their properties (oxidation state, hydrophylic properties) are compared to the measurements performed during the intensive measurement period of the summer 2013. In the simulation presented here, the ELVOC yield is assumed to be 11%, as detailed in section 2.2. Two sensitivity simulations are performed using a lower bound yield (3%) and an upper bound yield (18%). In Appendix D, similarly to what is presented in this section for the reference simulation, the sensitivity simulations are compared to each other and to the measurements in terms of the mass of $OM_1$, the organic aerosol composition, the OM:OC and O:C ratios.

### 4.1 Organic concentrations

The comparisons of the measured and modeled temporal profiles of the concentration of the submicron organic mass ($OM_1$) at Ersa are shown in Figure [3] for the two summer campaigns of 2012 and 2013.

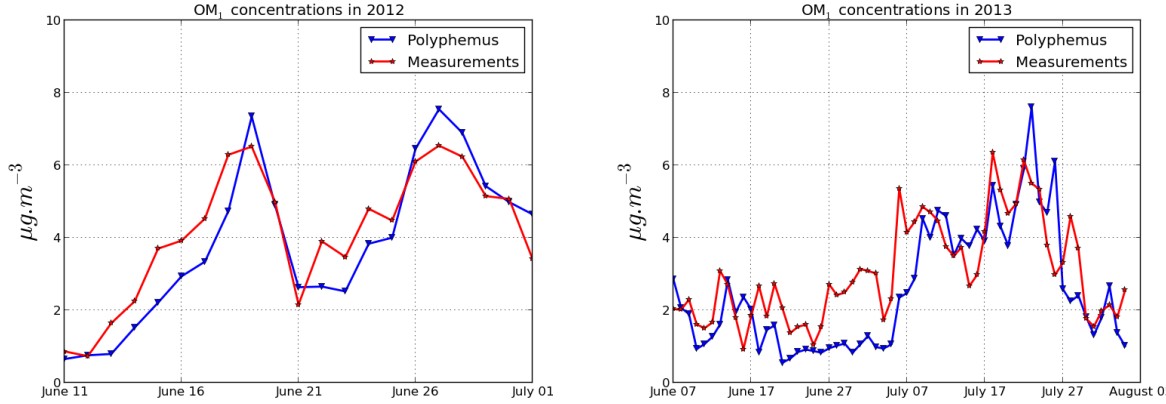

**Figure 3.** Comparison of measured and simulated daily $OM_1$ concentration at Ersa during the summer campaigns of 2012 (left panel) and 2013 (right panel).

The model shows satisfactory results at Ersa station, as shown by the statistics in Table [4]. Both the goal and the performance criteria of Boylan and Russell (2006) are verified for both years (MFB < 30% and MFE < 50%). The overall concentration of $OM_1$ is reasonably well modeled, although it is slightly underestimated 2.58 (respectively 3.71) $\mu$g.m$^{-3}$ against 2.89 (respectively 4.14) $\mu$g.m$^{-3}$ in the measurements for 2013 (respectively 2012). Overall, the model reproduces very well the peaks and troughs of $OM_1$ concentrations in both 2012 and 2013 with the exception of a few days in late June-early July 2013. This period is a period with air trajectories passing over France (Arndt et al., 2017), during which aging processes of biogenic compounds may be particularly important with formation of aged hydrophilic SOA. In the model, this process would lead to the formation of aged carboxylic acids, with MBTCA used as surrogate. However, the simulated concentration is low because the yield used is very low (it corresponds to the yield of MBTCA only).

| Year | $\bar{o}$ [$\mu$g.m$^{-3}$] | $\bar{s}$ [$\mu$g.m$^{-3}$] | RMSE [$\mu$g.m$^{-3}$] | Correlation [%] | MFB | MFE |
|------|------|------|------|------|------|------|
| 2012 | 4.14 | 3.71 | 2.00 | 61.7 | -0.13 | 0.39 |
| 2013 | 2.89 | 2.58 | 1.53 | 67.3 | -0.15 | 0.49 |

**Table 4.** Statistics of model to measurements comparisons for hourly organic concentrations in particles of diameters lower than 1 $\mu m$ during the summer campaigns of 2012 and 2013. $\bar{o}$ and $\bar{s}$ are the observed and simulated means, respectively. RMSE is the root mean square error, MFB and MFE are the mean fractional bias and error, respectively (see AppendixC).

## 4.2 Sources of OA

In both 2012 (6 June to 8 July) and 2013 (6 June to 10 August), the modeled organic mass is dominated essentially by biogenic particles (figure[4]). They represent 77% and 75% of the organic mass. In the model, for comparisons to the [14]C measurements, the biogenic-origin organic compounds are assumed to be non fossil, and the anthropogenic-origin organic compounds are assumed to be fossil. Although in winter, some of the anthropogenic-origin organic compounds may originate from wood combustion (residential heating) and be non fossil, we assume that the fraction of anthropogenic-origin organic compounds from residential heating is low in summer. The simulated OC is computed by dividing the modeled organic mass of each model surrogate by the OM/OC ratio of the surrogate.

During the period when [14]C measurements were performed (16 to 30 July), 75% of the modeled organic mass is biogenic, in agreement with the [14]C measurements, which estimated that 85% is non fossil. The measured and simulated means are 2.5 $\mu$gC.m$^{-3}$ and 1.9 $\mu$gC.m$^{-3}$ respectively for non-fossil OC, and 0.5 $\mu$gC.m$^{-3}$ and 0.6 $\mu$gC.m$^{-3}$ respectively for fossil OC. Although fossil OC is well modeled, non-fossil OC is slightly under-estimated between 16 and 30 July.

The modeled average chemical compositions of OM$_1$ at Ersa during the summers 2012 and 2013 are presented in Figure [4]. The chemical composition is very similar between the years 2012 and 2013.

Monoterpene oxidation products including ELVOCs and organic nitrate represent a large part of biogenic aerosols (about 48% in 2012 and 2013). ELVOCs and organic nitrate are abundant. ELVOCs represent 10% of OM$_1$ in 2012 and 15% in 2013. Organic nitrate represent 24% of the organic mass in 2012 and 20% in 2013. The route to organic particulate nitrate may essentially (but not exclusively) be active during the night as NO$_3$ efficiently photolyzes during the day and the production yields are more important during the night, and higher organic-nitrate concentrations are observed at night (Figure 5).

MBTCA, an oxidation product of monoterpenes, represents a tiny portion of OM$_1$, following the very low molar yield used. After monoterpenes, the most important biogenic precursor is isoprene; its oxidation products represent about 20% of OM$_1$ in 2012 and 16% in 2013. Although sesquiterpenes emissions are lower than isoprene and monoterpenes emissions, their oxidation products represent about 10% of OM$_1$. Anthropogenic oxidation products represent about 22% and 25% of OM$_1$ in 2012 and 2013 respectively. Most of anthropogenic oxidation products originate from intermediate and semi-volatile organic emissions (about 19% of OM$_1$, they are referred as anthropogenic SOA and POA in Figure [4]), and from aromatic oxidation products (3 to 5% of OM$_1$).

## 4.3 Oxidation state of organics

The level of oxidation of ambient organic aerosols is assessed by the organic matter to organic carbon ratio (OM/OC) and the oxygen-to-carbon ratio (O:C). OM is made up of many different molecular structures and it may include not only particulate organic carbon but also oxygen, hydrogen, nitrogen and/or sulphate. Hence, a high OM/OC ratio indicates a high degree of oxidation of the organic aerosols, and probably a high degree of hygroscopicity (Jimenez et al., 2009). There is a variety of methods that have been used to calculate OM/OC ratio as reported by Xing et al. (2013). In our case, the ambient OM/OC ratio is calculated by weighting the ratio $(OM/OC)_i$ of each surrogate species $i$ by the relative mass of the surrogate: $OM/OC =$

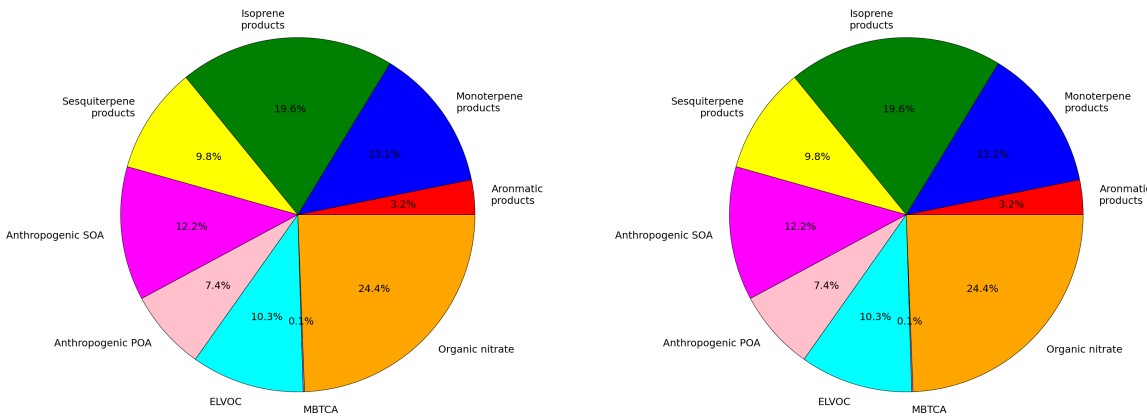

**Figure 4.** Simulated composition of $OM_1$ during the summer campaigns of 2012 (left panel) and 2013 (right panel).

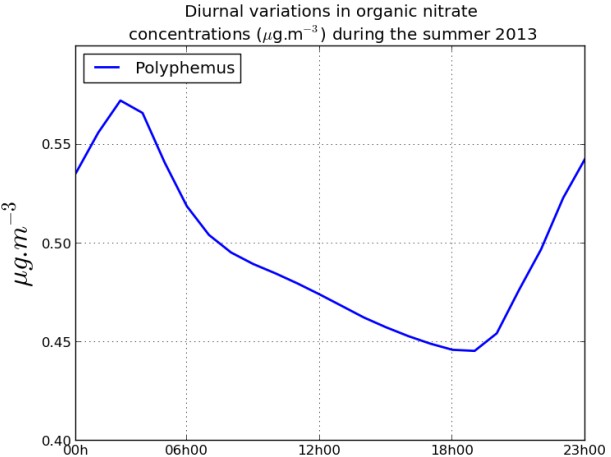

**Figure 5.** Diurnal variations in simulated organic nitrate concentrations during the summer 2013 at ERSA.

$\sum_{i=1}^{N_{esp}} (OM/OC)_i \times OM_i/OM$, where $Nesp$ is the number of surrogate species. The ratio $(OM/OC)_i$ of the surrogate species $i$ depends only on the molecular structure of the species and the number of carbons in the molecule. The O:C oxygen to carbon ratio allows for the degree of oxygenation of the organics to also be considered.

The measured and simulated temporal evolutions of both OM/OC and O:C ratios for submicron organic aerosols are shown in Figure [6] during the whole summer 2013 campaign period. The contributions from ELVOCs, organic nitrate and MBTCA are highlighted.

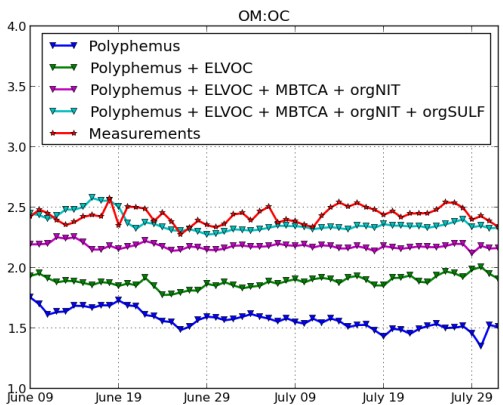 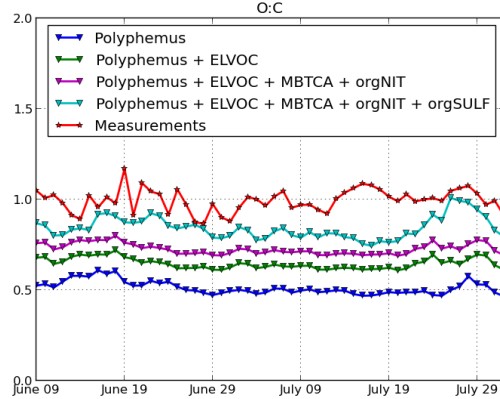

**Figure 6.** Daily variations of the ratios OM/OC (left panel) and O:C (right panel) during the 2013 campaign. The red line represents the measurements. The blue line represents model results without taking into account the concentrations of ELVOCs, MBTCA and organic nitrate. The green and magenta lines represent the model results by also taking into account ELVOCs (green) and MBTCA and organic nitrate (magenta). The cyan line represents the model results when assuming that a surrogate from isoprene oxidation is an organo-sulfate.

Relying on the measured values, the organic species over Ersa are highly oxidised and oxygenated. In fact, the measured value of the OM/OC ratio (2.43) is higher than the value of 2.1 suggested by Turpin and Lim (2001) for a rural site like Kern Refuge (U.S.). The measured O:C ratio is 0.99. In agreement with the measurements, all simulations show a relatively stable OM/OC ratio and O:C ratio during the simulated period.

5 Without taking into account the ELVOCs, MBTCA and organic nitrate species, the model strongly under-estimates both the OM/OC ratio and the O:C ratio. This is because of the absence of highly oxidized species in the model, as all the other modeled organic compounds with non-negligible mass tend to have low OM/OC and O:C ratios.

Taking into account the formation of ELVOCs leads to improvements in the predicted oxidation state of aerosols: the OM/OC ratio (respectively O:C) changes from 1.57 to 1.89 (respectively 0.51 to 0.65), although the monomer and dimer ELVOCs only 10 represent 15.7% of the $OM_1$ mass.

MBTCA has a low impact on the organics oxidation level, despite its high OM/OC ratio, because it constitutes only a tiny part of the $OM_1$ mass (0.2%).

Taking into account organic nitrate leads to an improvement of both ratios. The OM/OC increases from 1.89 to 2.18 and the O:C ratio increases from 0.65 to 0.73.

15 A possible way to explain the under-estimation of the OM/OC ratio (2.17 simulated against 2.43 measured) and the O:C ratio (0.73 simulated and 0.99 measured) is to take into account the formation of organo-sulfate. As both organic and sulfate are the major components of aerosols at Ersa (Nicolas, 2013), there may be formation of organo-sulfate, as suggested by the transmission electron microscopy measurements of Freney et al. (2017) in the South of France.

The measurements performed at Ersa show a good correlation between sulfate and organic $OM_1$ concentrations, with a linear regression coefficient of 0.64. In agreement with the measurements, the simulated concentrations of sulfate and organics are also well correlated with a linear regression coefficient of 0.71. Although the formation of organo-sulfate is not modeled here, the modeled correlation is high because both sulfate and organics are formed by oxidation of precursors, and oxidant concentrations largely depend on meteorological variables, such as temperature and dilution within a variable mixing layer. Furthermore, a large part of biogenic SOA is hydrophilic and therefore higher condensation of sulfate enhances their partitioning into the particulate phase, as the mass of the aqueous phase increases through the condensation of sulfate (Couvidat and Sartelet, 2015).

In laboratory, the formation of organosulfate was observed from the uptake of monoterpene oxidation products (pinonaldehyde) on acidic sulfate aerosols (Liggio and Li, 2006; Surratt et al., 2008), from the uptake of ELVOC (Mutzel et al., 2015), and from the uptake of isoprene oxidation products (Liggio et al., 2005; Nguyen et al., 2014). Isoprene SOA may be formed via the reactive uptake of isoprene-epoxydiol (IEPOX), a second generation oxidation product of isoprene, in the presence of hydrated sulfate (Surratt et al., 2010; Couvidat et al., 2013b; Nguyen et al., 2014). Using aerosol mass spectrometer measurements, Hu et al. (2015) estimated that IEPOX-OA makes a large fraction of the OA (between 17 to 36% in the U.S.) outside urban areas, in agreement with Budisulistiorini et al. (2015). In regions where aerosols are acidic, IEPOX-derived OA may be strongly dependent on the sulfate concentration, which acts as nucleophile and facilitates the ring-opening reaction of IEPOX and organosulfate formation (Nguyen et al., 2014; Xu et al., 2015).

In order to take into account the influence of the formation of organo-sulfates on OA properties, the surrogate products of the model are modified. As Mediterranean aerosol composition displays large concentrations of sulfate, isoprene oxidation products may lead to the formation of organo-sulfate. In the model, the components formed from the low-NOx oxidation of isoprene are BiPER and BiDER. The surrogate BiPER is supposed to be a methyl dihydroxy dihydroperoxide. Although BiDER is not identified, it is assumed to have the properties of a methyl tetrol (Couvidat and Seigneur, 2011). If we assumed that this compound has the same properties as a sulfate ester of formula $C_6H_{11}O_3SO_4$, the ratios OM/OC and O:C increase to get closer to measurements (Figure 6). In fact, the average OM/OC ratio (respectively O:C) increases from 2.18 (respectively 0.73) to 2.37 (respectively 0.84), which compares very well with the average measured ratios (2.43 for OM/OC and 0.99 for O:C). Even though the ratio O:C still seems to be slightly under-estimated, the discrepancies may be explained by uncertainties in the measurements. Measurements performed at the same place between 10 July and 6 August with an HR-toF-AMS (aerosol mass spectrometer) shown an average OM/OC ratio of 2.34 ($\pm$0.14) and an average O:C ratio 0.92 ($\pm$0.11).

## 4.4 Water-soluble organics

Water-soluble organics constitue a major fraction of organic compounds. On average between 15 and 31 July 2013, submicron water-soluble organic carbon (WSOC) represents 64% of the organic carbon in the measurements and 46% in the model. WSOC concentrations are well modeled on average (the measured mean is 1.0 $\mu$gC.m$^{-3}$ and the modeled mean is 0.9 $\mu$gC.m$^{-3}$). Figure [7] shows the daily concentrations of WSOC in the model and according to the measurements. Although the WSOC concentrations are well modeled between 21 and 31 July, they are under-estimated between 15 and 20 July. These

differences between the periods in the ability of the model to represent WSOC concentrations may be linked to differences in the organic aerosol composition and in the origins of air masses. This is illustrated by the comparison of 16 and 30 July. WSOC concentrations are under-estimated on 16 July, but are well modeled on 30 July. 16 July is characterised by low/calm winds in a time when 30 July is characterised by winds from south-east France. Figure [8] shows the chemical composition of modeled OM$_1$ for both days. On 30 July, hydrophilic oxidation products of isoprene constitute most of the mass: they represent 26% of the concentrations against 9% on 16 July. However, organic nitrate (from monoterpene oxidation) represents as much as 35% of OM$_1$ concentrations on 16 July, against only about 10% on 30 July, because the low winds of 16 July probably enhance the influence of NO$_x$ emissions from ships on pollutant concentrations.

Although this organic nitrate is assumed to be hydrophobic, it may have undergone hydrolysis resulting in nitric acid and nonvolatile secondary organic aerosol that may change the hydrophylicity of organics and the organic composition, as detailed in Pye et al. (2015).

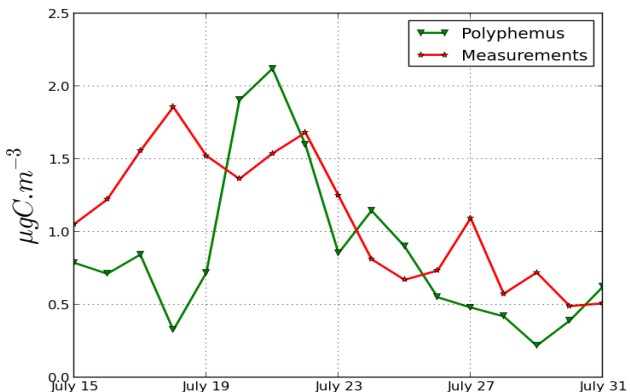

**Figure 7.** Measured and simulated submicron water soluble organic carbon ($\mu$gC.m$^{-3}$) at Ersa

## 5   Impact of the biological activity of the Mediterranean Sea

According to O'Dowd et al. (2004), organic aerosols of marine origin can contribute to organic OM$_1$ concentrations especially near biologically productive waters. Particles of diameters larger than 1 $\mu$m tend to contain mostly inorganic compounds, and the fraction of organic increases with decreasing diameter for particles of diameters smaller than 1 $\mu$m (O'Dowd et al., 2004; Schwier et al., 2015).

Several studies found a correlation between the organic mass fraction of sea-spray aerosol (OM$_{SSA}$) and the concentrations of oceanic parameters like chlorophyll-a [chl-a], which is used as a proxy for biological activity and related ocean chemistry (O'Dowd et al., 2008; Gantt et al., 2011; Schwier et al., 2015). Several parameterisations exist to estimate the fraction of organics in primary marine aerosols. Whereas in the parameterisation of Schwier et al. (2015), which is designed for Aitken

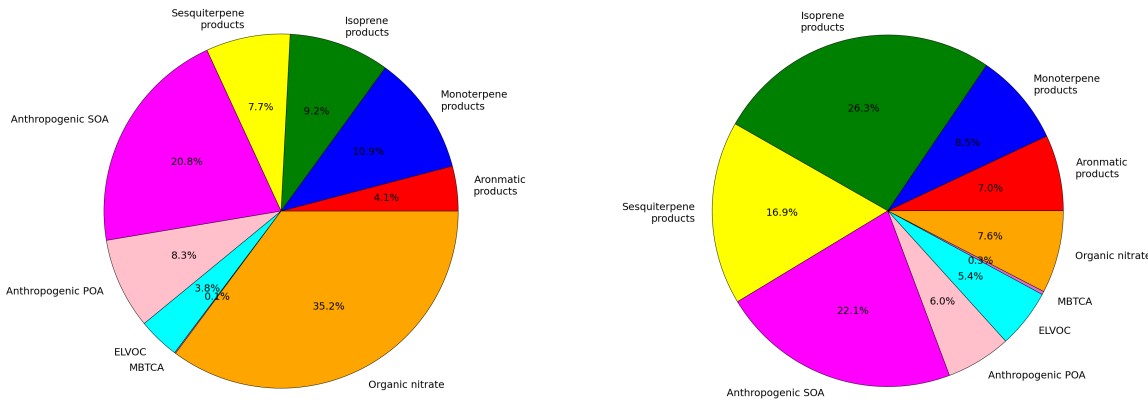

**Figure 8.** Simulated chemical composition of $OM_1$ on 16 July (left panel) and on 30 July 2013 (right panel)

mode aerosols, the organic fraction only depends on chl-a, it depends also on the 10 m wind speed and the particle diameter in Gantt et al. (2012). $OM_{SSA}$ decreases with increasing 10 m wind speed, because for strong wind speeds, bubbles are not enough enriched by organic matter due to the wave breaking. Concerning the size-dependence, Gantt et al. (2012) showed that the maximum organic fraction in the Aitken and accumulation modes is about 80 to 90%, while the fraction is less than 2% in

5 the coarse mode.

The concentrations of the chlorophyll-a are obtained from monthly averaged MODIS/AQUA satellite data (http://oceandata.-sci.gsfc.nasa.gov/MODIS-Aqua/L3SMI, Hu et al. (2012)), with a spatial resolution of 4km×4km. They are shown in the left panel of Figure [9].

Values are low, typical of oligotrophic conditions that characterize stratified surface Mediterranean waters in summer. The

10 highest chl-a concentrations are recorded around the coastal zones meaning shallow water, places where sea currents bring cold waters with plants and nutrients from sea floor or brought from the rivers to the surface due to the rising slope of the sea floor.

Far from the coasts, the chl-a is more or less uniform (less than 0.2 mg.m$^{-3}$). The chl-a temporal average near Ersa is 0.14 mg.m$^{-3}$.

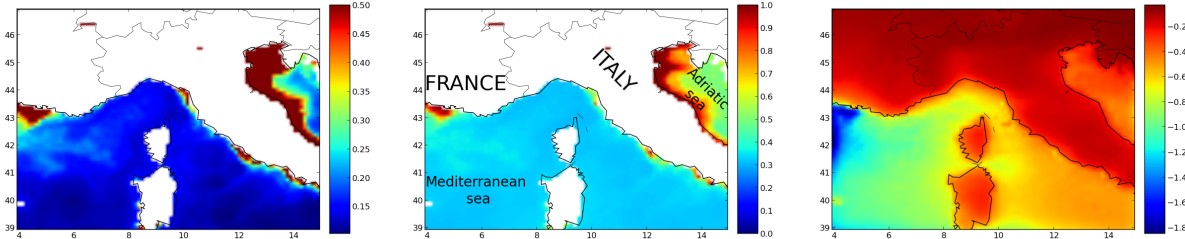

**Figure 9.** Left panel: Temporal average of chl-a in sea surface in units of $mg.m^{-3}$ during the summer of 2013 (from 6 June to 3 August). Middle panel: Organic mass fraction of emitted sea-spray aerosols of diameters between 0.01 and 0.1585 $\mu$m. Right panel: Relative difference (%) in the concentrations of $OM_1$ between the base simulation and the simulation including organic sea-salt emissions.

The emitted organic mass fraction of sea salt is shown in the middle panel of Figure [9] using the parameterisation of Gantt et al. (2012) for aerosols of diameters between 0.01 and 0.1585 $\mu$m. The organic fraction map has almost the same spatial distribution as the chl-a map. The fraction of marine organic emissions is higher near the shores of the continent. The temporal average of emitted $OM_{SSA}$ near Ersa is detailed in Table [5] for aerosols of different sizes. As expected, the organic fraction

5 is higher for aerosols in the Aitken mode ($\sim 0.31$) than in the accumulation mode ($\sim 0.22$) than in the coarse one ($\sim 0.01$). These simulated organic fractions are consistent with the fraction obtained by the Schwier et al. (2015) parameterisation for the Aitken mode (0.23). They are also consistent with the average fraction of 0.24 found experimentally by Schwier et al. (2015) using pelagic mesocosms in the bay of Calvi (Corsica, France) during the summer of 2012 (an average fraction of 0.24).

| Diameter range [$\mu$m] | [0.01,0.1585] | [0.1585,1.0] | [1.0,2.5119] | [2.5119,10.0] |
|---|---|---|---|---|
| Temporal average of $OM_{SSA}$ in Ersa | 0.31 | 0.22 | 0.02 | 0.01 |

**Table 5.** Temporal averages of $OM_{SSA}$ near Ersa using the parameterisation of Gantt et al. (2012) as a function of aerosol particle diameter, over the summer of 2013 (from 6 June to 3 August).

A simulation was performed using the organic fraction of primary marine emissions computed with the Gantt et al. (2012) parameterisation, in order to assess the impact of the marine organics on the concentrations of organics $OM_1$. The organic emissions are affected to a new species called "SSorg", which is assumed to be hydrophobic and not very volatile. The properties are detailed in Table [3]. The values are assumed to be the same as those taken for methyl nitro benzoic acid in the model, following the lack of data characterising the properties of these species. Secondary marine OM formation is not taken into account.

The right panel of Figure [9] shows the relative difference of organic $OM_1$ concentrations between the base simulation and the simulation including organic sea-salt. The contribution of the organic sea-salt emissions to organic $OM_1$ is small (a few percent at the maximum) and localised above the sea. On average over the marine domain, the organic sea-salt emissions contribute to about 1.8% of the organic $OM_1$ concentrations. Despite the larger chl-a and organic fraction over the Adriatic Sea, the contribution of SSorg to $OM_1$ concentrations is not as high as the one over the Mediterranean Sea in the South of France. This is due to the fact that the surface wind-driven flux of sea salts over the Adriatic Sea is not as important as over the Mediterranean Sea in the South of France.

## 6   Conclusion

This paper presents comparisons of modeled organic concentrations and properties to surface measurements performed at Ersa (Cap Corsica, France) during the summers 2012 and 2013. The air-quality model of the Polyphemus platform is used with a surrogate approach to model secondary organic aerosols (SOA). The previously-published surrogate approach is modified to better represent observed organic aerosol (OA) properties, by taking into account the formation of extremely low volatility organic compounds (ELVOCs) and organic nitrate from monoterpene oxidation. The concentrations of organic matter compare well to the measurements performed with an ACSM (Aerosol Chemical Speciation Monitor).

During the summer 2013, the added surrogates led to a significant increase of mass concentrations: they contributed to 15% of the $OM_1$ mass for ELVOC, 20% for organic nitrate from monoterpene oxydation and 0.2% for MBTCA. In agreement with [14]C measurements, most of the organic aerosol is from non-fossil (biogenic) origin. The inclusion in the model of ELVOC and organic nitrate formation greatly improves the modeled oxidation state of particles, as assessed by the OM/OC and O:C ratios. Despite the model improvements, these ratios remain under-estimated compared to measurements (2.18 simulated against 2.43 measured for OM/OC and 0.73 simulated against 0.99 measured for O:C). However, the ratios are better modeled by assuming that a surrogate species from isoprene oxidation is an organo-sulfate (2.37 simulated for OM/OC and 0.84 for O:C), suggesting that further work should focus on a better description of organo-sulfate formation. Although an organic acid, MBTCA, was introduced as a second generation product of monoterpenes, its yield should be revisited to include the formation of several carboxylic acids, rather than a single species. Concerning the hydrophilic properties of aerosols, as much as 64% of organic carbon is soluble. Therefore, taking into account the hydrophylic properties of aerosols is crucial to model the partitioning of aerosols between the gas and particle phases. The average concentration of water soluble organic carbon is relatively well modeled by comparison to measurements performed using a PILS-TOC-UV (with a mean value of 1.0 $\mu$gC.m$^{-3}$ in the

measurements and 0.9 $\mu$gC.m$^{-3}$ in the model) over the second half of July 2013. Daily comparisons to measurements show that although organic nitrate is assumed hydrophobic here, its hydrolysis should be modeled to better represent the hydrophylic properties of organics. There are other pathways and mechanisms that are not considered in the model, but that may change the concentrations and hydrophylicity of organics (Shrivastava et al., 2017). For example, salting effects (via activity coefficients)

are not considered, although inorganics provide a mass onto which hydrophilic organic surrogates may condense. Furthermore, pathways such as the aging chemistry of VOCs and I/S-VOCs from biomass burning (wildfires) and organic cloud processing are not considered. However, these pathways may be relatively low during the studied periods. Marine organic aerosols were added to the model, with a parameterisation depending on chlorophyll-a, 10 m wind speed and particle diameters. Although the emitted organic fraction is high for particles of small diameters (Aitken and accumulation modes), its contribution to the

total organic mass OM$_1$ is only a few percents at the maximum. Its contribution over the continent is always lower than 1 to 2%.

*Acknowledgements.*  This research has received funding from the French National Research Agency (ANR) projects SAF-MED (grant ANR-12-BS06-0013). This work is part of the ChArMEx project supported by ADEME, CEA, CNRS-INSU and Météo-France through the multidisciplinary programme MISTRALS (Mediterranean Integrated Studies aT Regional And Local Scales). The station at Ersa was partly

supported by the CORSiCA project funded by the Collectivité Territoriale de Corse through the Fonds Européen de Développement Régional of the European Operational Program 2007-2013 and the Contrat de Plan Etat-Région. Eric Hamounou and François Dulac are acknowledged for their great help in organizing the campaigns at Ersa. CEREA is a member of Institut Pierre-Simon Laplace (IPSL).

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

## Appendix A: Formation of ELVOC

The reactions involved in the formation of ELVOCs are

$$MT + O_3 \quad \xrightarrow{\quad k_1 \quad} \quad \gamma_{RO_2}.RO_2 + (1 - \lambda - \gamma_{RO_2}).R'O_2 + OH + 0.001\ HO_2 \tag{A1}$$

$$R'O_2 \quad \xrightarrow{\quad k_{H/O2} \quad} \quad \alpha.R''O_2 \tag{A2}$$

$$R''O_2 \quad \xrightarrow{\quad k_{H/O2} \quad} \quad \alpha.R'''O_2 \tag{A3}$$

$$R'''O_2 \quad \xrightarrow{\quad k_{H/O2} \quad} \quad \alpha.R_{elvoc}O_2 \tag{A4}$$

$$RO_2 + R'O_2 \quad \xrightarrow{\quad k_2 \quad} \quad Products \tag{A5}$$

$$RO_2 + NO \quad \xrightarrow{\quad k_3 \quad} \quad \delta.RO_2 \tag{A6}$$

$$RO_2 + HO_2 \quad \xrightarrow{\quad k_4 \quad} \quad Products \tag{A7}$$

$$R_{elvoc}O_2 + RO_2(or\,HO_2) \quad \xrightarrow{\quad k_5 \quad} \quad \beta.Monomer + (1 - \beta).Dimer \tag{A8}$$

$$R_{elvoc}O_2 + NO \quad \xrightarrow{\quad k_6 \quad} \quad \chi.nitrate_{org} + \phi.Monomer \tag{A9}$$

| Kinetic constant | Value ($cm^3 s^{-1}$ or $s^{-1}$) |
|---|---|
| $k_1$ | $8.4\ 10^{-17}$ |
| $k_2$ | $1.0\ 10^{-12}$ |
| $k_3$ | $4.7\ 10^{-12}$ |
| $k_4$ | $2.7\ 10^{-11}$ |
| $k_5$ | $5.0\ 10^{-11}$ |
| $k_6$ | $4.7\ 10^{-12}$ |
| $k_{H/O_2}$ | $0.5$ |

**Table A1.** Kinetic constants used in the ELVOC kinetic model

After numerous combinations, the stoichiometric coefficients of equations [A1] were determined, such as reproducing the observations of Ehn et al. (2014) for both the low-NOx and high-NOx regimes (Table A2).

| Stoichiometric coefficient | $\gamma_{RO_2}$ | $\lambda$ | $\alpha$ | $\delta$ | $\beta$ | $\chi$ | $\phi$ |
|---|---|---|---|---|---|---|---|
| Value | 0.0002 | 0.0998 | 1.0 | 0.995 | 0.6 | 0.00012 | 0.00001 |

**Table A2.** Stoichiometric coefficients used in the model

For model validation, comparisons are made to the experiments of Ehn et al. (2014). In the low-NOx regime, the experiments of Ehn et al. (2014) lasted 45 min and the initial ozone concentration was 80 ppb. In the model, we modified the initial concentration of $\alpha$-pinene from 0 ppb to 11 ppb to reproduce the observations. Figure [A1], which shows one-fifth of the ELVOC concentrations as a function of the $\alpha$-pinene reaction rate, reproduces successfully the extended data Figure 10 of Ehn et al. (2014). According to Figure [A1], the increase of peroxy radicals has a square root dependence while ELVOC monomers and dimers evolve nearly linearly. Therefore, the total ELVOC concentration has a near-linear dependence on the amount of $\alpha$-pinene reacting with $O_3$ indicative of first-generation oxidation products. These findings are consistent qualitatively and quantitatively with Ehn et al. (2014) measured and modeled results.

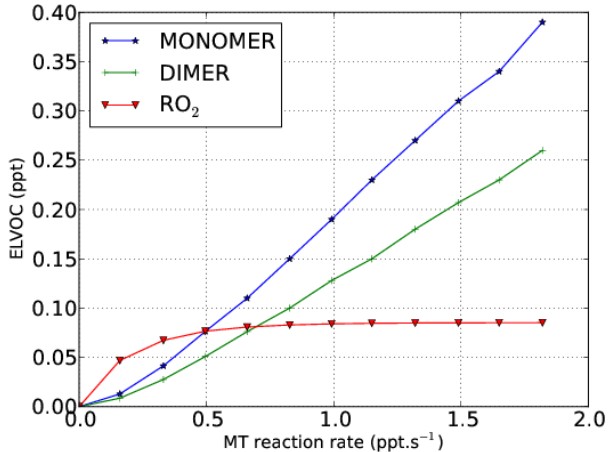

**Figure A1.** ELVOC concentrations as a function of the $\alpha$-pinene reaction rate for the low-$NO_x$ experiment

In the high $NO_x$ regime, the experiments of Ehn et al. (2014) also lasted 45 min and the initial ozone and $\alpha$-pinene concentrations were 80 ppb and 5 ppb respectively. The initial NO concentration was changed gradually from 0.3 ppb to 5 ppb. Figure [A2], which shows one-fifth of the ELVOC as a function of the $RO_2$ concentration, reproduces successfully the Extended Data Figure 10 of Ehn et al. (2014). While increasing NO concentration, both monomer and dimer concentrations decrease rapidly as expected because a fraction of peroxy radicals is consumed by the NO reaction. Moreover, the dimer concentration decreases rapidly while the monomer concentration decreases more slowly. Simultaneously, the organic nitrate concentration increases with increasing NO.

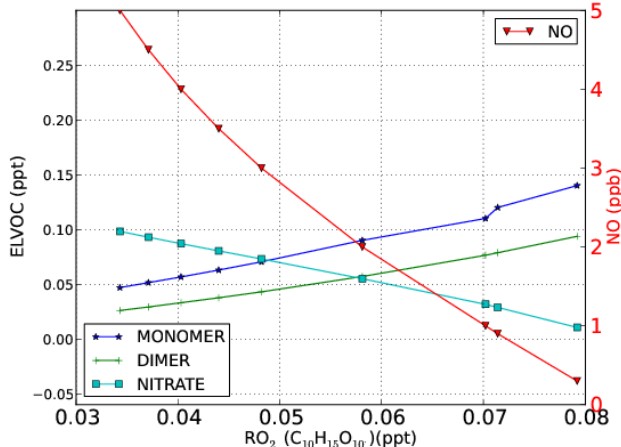

**Figure A2.** ELVOC concentration as a function of the $RO_2$ concentration in the high-$NO_x$ regime.

## Appendix B: Formation of organic nitrate

The formation of organic nitrate (orgNIT) from monoterpenes (MT) is modeled with the following reactions:

$$Terpene + OH \xrightarrow{k'_1} TERPRO_2 \tag{B1}$$

$$TERPRO_2 + NO \xrightarrow{k'_2} 0.201 orgNIT \tag{B2}$$

$$MT + NO_3 \xrightarrow{k'_3} TERPNRO_2 \tag{B3}$$

$$TERPNRO_2 + NO \xrightarrow{k'_4} 0.688 orgNIT + Products \tag{B4}$$

$$TERPNRO_2 + HO_2 \xrightarrow{k'_5} orgNIT + Products \tag{B5}$$

$$TERPNRO_2 + NO_3 \xrightarrow{k'_6} 0.422 orgNIT + Products \tag{B6}$$

$$TERPNRO_2 + RO_2 \xrightarrow{k'_7} (1 - 0.578\delta) orgNIT + Products \qquad \text{where } \delta = 50\% \tag{B7}$$

| Kinetic constant | Value ($cm^3 s^{-1}$ or $s^{-1}$) |
|---|---|
| k'$_1$ | $\alpha$-pinene ( 1.21 $10^{-11}$ exp(444/T)) |
| | Limonene (4.20 $10^{-11}$ exp(401/T)) |
| | $\beta$-pinene ( 2.38 $10^{-11}$ exp(357/T)) |
| | Humulene ( 2.93 $10^{-10}$) |
| k'$_2$ | 2.27 $10^{-11}$ exp(435/T) |
| k'$_3$ | $\alpha$-pinene ( 1.19 $10^{-12}$ exp(490/T)) |
| | Limonene (1.22 $10^{-11}$ ) |
| | $\beta$-pinene ( 2.51 $10^{-12}$) |
| | Humulene ( 1.33 $10^{-12}$ exp(490/T)) |
| k'$_4$ | 2.6 $10^{-12}$ exp(380/T) |
| k'$_5$ | 2.65 $10^{-13}$ exp(1300/T) |
| k'$_6$ | 2.3 $10^{-12}$ |
| k'$_7$ | 3.5 $10^{-14}$ |

**Table B1.** Kinetic constants used in the organic nitrate formation mechanism

## Appendix C: Statistic indicators and criteria

The statistic indicators used in this paper are described in Table C1. The performance and goal criteria used in this paper are described in Table C2.

| Statistic indicator | Definition |
|---|---|
| Root mean square error (RMSE) | $\sqrt{\frac{1}{n}\sum_{i=1}^{n}(c_i - o_i)^2}$ |
| Correlation (Corr) | $\dfrac{\sum_{i=1}^{n}(c_i - \bar{c})(o_i - \bar{o})}{\sqrt{\sum_{i=1}^{n}(c_i - \bar{c})^2}\sqrt{\sum_{i=1}^{n}(o_i - \bar{o})^2}}$ |
| Mean fractional bias (MFB) | $\frac{1}{n}\sum_{i=1}^{n}\dfrac{c_i - o_i}{(c_i + o_i)/2}$ |
| Mean fractional error (MFE) | $\frac{1}{n}\sum_{i=1}^{n}\dfrac{\mid c_i - o_i \mid}{(c_i + o_i)/2}$ |

**Table C1.** Definitions of the statistics used in this work. $(o_i)_i$ and $(c_i)_i$ are the observed and the simulated concentrations at time and location i, respectively. $n$ is the number of data

| Criteria | Performance criterion | Goal criterion |
|----------|----------------------|----------------|
| MFB | $\leq 60\%$ | $\leq 30\%$ |
| MFE | $\leq 75\%$ | $\leq 50\%$ |

**Table C2.** Boylan and Russel criteria

## Appendix D: Sensitivity to ELVOC yield

The ELVOC yield in the reference simulation is 11%. Two sensitivity simulations are performed, using a lower bound (3%, Jokinen et al. (2015)) and a higher bound (18%, Ehn et al. (2014)).

The comparison of $OM_1$ concentrations is shown in Figure D1 and the statistical evaluation is shown in Table D1. The correlation between the measurements and the simulation is not modified by the ELVOC yield. However, the higher the yield, the closer to zero the bias is (it decreases from -26% to -7%) and the lower the error MFE. However, the lower RMSE is obtained with a yield of 11% (1.53 $\mu$g.m$^{-3}$ with a yield of 11% against 1.54 $\mu$g.m$^{-3}$ with a yield of 3% and 1.59 $\mu$g.m$^{-3}$ with a yield of 18%).

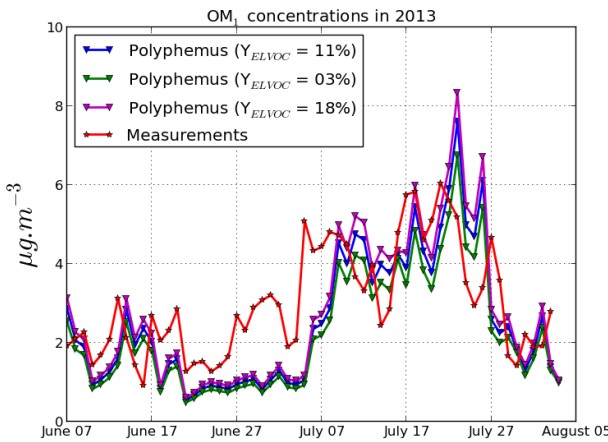

**Figure D1.** Simulated daily concentrations of $OM_1$ using a molar ELVOC yield of 11% (blue plot, reference simulation), 18% (magenta plot) and 3% (green plot).

| ELVOC yield (%) | $\bar{s}$ [$\mu$g.m$^{-3}$] | RMSE [$\mu$g.m$^{-3}$] | Correlation [%] | MFB | MFE |
|---|---|---|---|---|---|
| 03 | 2.30 | 1.54 | 67.2 | -0.26 | 0.52 |
| 11 | 2.58 | 1.53 | 67.3 | -0.15 | 0.49 |
| 18 | 2.82 | 1.59 | 67.2 | -0.07 | 0.47 |

**Table D1.** Statistics of model to measurements comparisons for organic concentrations in particles of diameters lower than 1 $\mu m$ during the summer campaign of 2013 using an ELVOC molar yield of 3%, 11% (reference) and 18%. $\bar{s}$ represents the simulated mean concentrations. The observed mean concentration is $\bar{o} = 2.89$ $\mu$g.m$^{-3}$.

The simulated composition of OM$_1$ using ELVOC yields of 3% and 18% are shown in Figure D2. Using an ELVOC yield of 3% (respectively 18%), the ELVOCs represent 4.7% (respectively 22.9%) of the OM$_1$ mass.

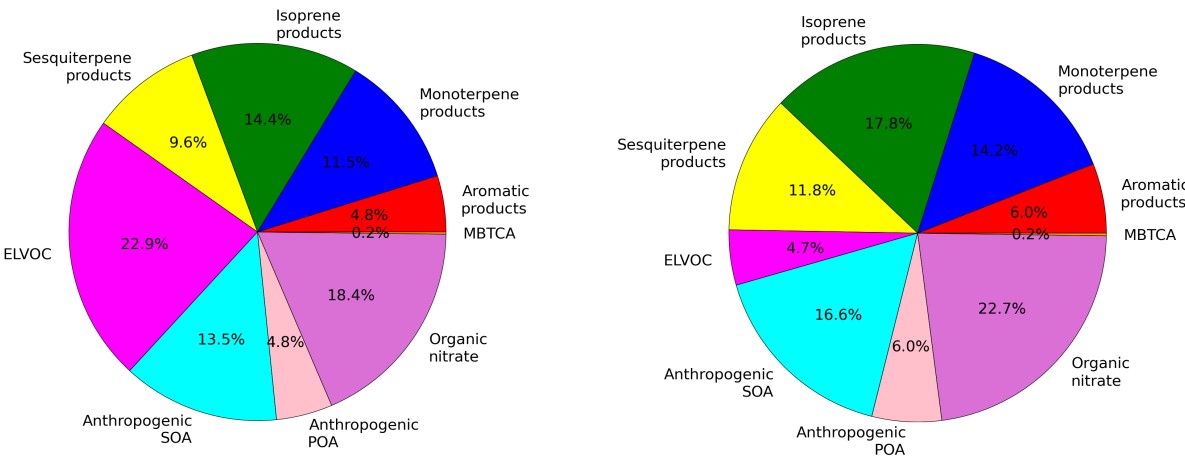

**Figure D2.** Simulated composition of OM$_1$ using the upper (left panel) and lower (right panel) bounds of ELVOCs molar yields

The OM:OC and O:C ratios are plotted using the three yields (3%, 11% and 18%) in Figure D3. The simulated means of the two ratios using the three ELVOC yields (3%, 11% and 18%) are shown in table D2. The ratios using the upper bound of ELVOC yields are the closest to the measurements. The OM:OC and O:C ratios may be higher if organo-sulfate are considered (section 4.3).

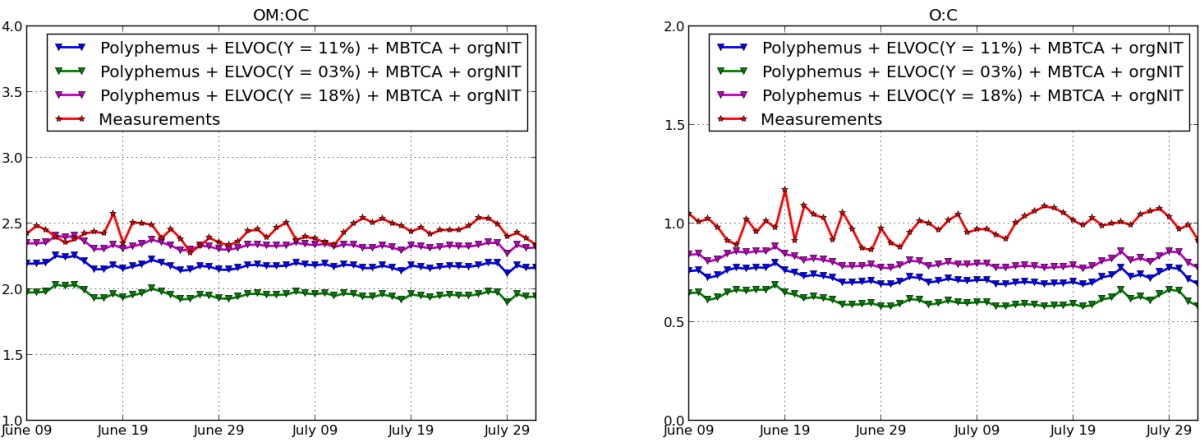

**Figure D3.** Comparisons of the OM:OC ratio (left panel) and the O:C ratio (right panel) for simulations using an ELVOC molar yield of 3%, 11% and 18%.

| ELVOC yield (%) | OM:OC | O:C |
|---|---|---|
| 3 | 1.96 | 0.61 |
| 11 | 2.18 | 0.73 |
| 18 | 2.33 | 0.81 |

**Table D2.** Simulated means of the OM:OC and O:C ratios during the summer campaign of 2013 using an ELVOC molar yield of 3%, 11% and 18%. The measured means of OM:OC and O:C are 2.43 and 0.99 respectively.