# Peer review of "Modelling organic aerosol concentrations and properties during ChArMEx summer campaigns of 2012 and 2013 in the western Mediterranean region"

_Atmospheric Chemistry and Physics, 2017_

## Referee Comment (RC1) · Anonymous Referee #3 · 3 Jul 2017

Chrit et al. have performed simulations using an air quality model and compared predictions of organic aerosol mass and composition to measurements at a remote site in the Mediterranean Sea. They found that model updates based on the inclusion of new secondary organic aerosol formation pathways improved the model-measurement comparison. Overall the manuscript is well motivated, researched, and discussed. I recommend publication after the authors have had an opportunity to respond to my comments and considered my suggestions.

1. Page 1, line 5: Consider using the noun form: hydrophilicity.

[Figure]

2. Page 1, line 15: 'percent' not 'percents'.

3. Page 2, lines 1-10: References are dated. Consider newer references.

4. Page 2, lines 8-9: Rephrase.

5. Page 2, lines 13: Intermediate-volatility organic compounds are a separate precursor category. Add discussion in the introduction.

6. Page 2, lines 16-22: It would be helpful to be more quantitative when citing earlier work. For example, what fraction of the organic aerosol that El Haddad et al. (2011, 2013) measured was biogenic in nature?

7. Page 2, lines 17-19: Hayes et al. (2013, 2015) argue that the biogenic SOA found in Los Angeles was produced near the source and then transported into the city. Suggest citing and reconciling with Hayes work.

8. Page 2, line 20, Page 9, line23 (and elsewhere too): 'Fossil' and not 'fossile'.

9. Page 2, line 19: Was Minguillon a measurement or modeling study?

10. Page 3, line 16: Incomplete sentence: 'monoterpenes oxidation products SOA over the U.S.'.

11. Page 4, Section 2.1: Can you briefly summarize the existing SOA precursors, species, and processes in the model and discuss how those earlier processes do not overlap with the updates made in this work?

12. Page 5, line 26-28: Am I correct that these oxygenated peroxy radicals can be formed only in the absence of NO? If yes, specify.

13. How is gas/particle partitioning of the explicit oxidation products modeled? If only briefly, please summarize the partitioning model and assumptions.

14. Page 8, line 2: I did not understand the meaning of 'nested rep.' in parentheses.

15. Page 8, line 14: 'split' not 'splitted'.

16. Page 8, line 18-20: The SVOC/POA ratios might be a little too high. See work of May et al. (2014a,b) for estimates on SVOC/POA ratios. Also, is there a reason why IVOC emissions were not considered? See work of Jathar et al. (2014) and Zhao et al. (2015, 2106) for estimates of IVOCs from combustion sources as a function of VOC emissions.

17. Page 8, lines 25-28: What is the organic fraction in sea salt emissions?

18. Page 10: How does the model perform in predicting the diurnal variations in OA?

19. Figure 2: The composition in Figure 3 tells me that the model updates must also have increased mass concentrations and resulted in better comparison in Figure 2. This fact is missing in Section 4.1. Please state the importance of this either in Section 4.1 or in the conclusions in Section 6.

20. Section 4.2: Has the model been evaluated for other pollutants? For example, black carbon, carbon monoxide, ozone, NO.

21. Page 11, line 20: How low are the sesquiterpene emissions compared to isoprene and monoterpenes? Be quantitative.

22. Page 11, line 22: IVOCs are mentioned here but are they actually included since the methods do not talk about them.

23. Figure 3 and 6, left panel: Why is the pie not a circle? Also, consider increasing the font size for readability.

24. Page 12, line 5: 'There are a variety...'.

25. Page 12, line 7: 'ponderating'?

26. Figure 4: The legend makes it seem like the model predictions systematically discount (or subtract) the effects of the three updates rather than the opposite that is mentioned in the caption. Replace '-' with '+'?

27. Figure 4: Given that the OM:OC and O:C measurements are not directly measured but rather interpreted from the ACSM data, can the measurements be shown with error bars?

28. Page 14, lines 5-7: Can the authors describe the mechanism at play here and the correct citation?

29. Section 4.4 and Page 1, line 11: The claims about improving the hydrophilicity predictions are slightly misleading since what the authors have actually done is improve predictions of mass concentrations of water soluble organic carbon.

30. Figure 6 does not have a left-right panel but a top-bottom panel. Fix caption.

31. Figures 7, 8 and 9 could be combined into a single multi-panel figure. Also, consider adding city names to orient the reader not familiar with that part of the world.

32. Sensitivity analysis: How sensitive are the model predictions and the findings from this work to the various inputs (reaction rate constants, yields, etc.) listed in the appendix? I would encourage the authors to perform additional simulations to (i) develop lower and upper bounds on their estimates for ELVOCs, organic nitrates, and MBTCA and (ii) develop insight on the most important inputs that would guide future laboratory work.

---

## Referee Comment (RC2) · Anonymous Referee #1 · 13 Jul 2017

Review of Chrit et al. This paper models the organic aerosol concentration and properties during the ChArMEx summer campaigns of 2012 and 2013 using a surrogate approach for SOA formation. Although I agree with the importance of this topic and its novelty, the modeling approaches need to clearly explained. Also, the caveats of these approaches need to better acknowledged. Below are my specific comments that should be addressed before the paper is accepted for publication. 1. The abstract mentions that surrogate species for monoterpenes are a-pinene and limonene. But on page 5, the authors mention three precursors: a-pinene, b-pinene and limonene. This

needs to be clarified. Also, why were these 2 particular species chosen as surrogates for monoterpenes?

2. Page 3: Why was IEPOX not observed during ChArMex campaign? Did the instruments measure gas-phase IEPOX or particle phase iepox components? Was the aerosol acidic or neutral?

3. The authors mention the importance of organosulfates in this study. But IEPOX chemistry also makes organosulfates. If these are not coming from isoprene or IEPOX chemistry, what are alternate mechanisms for organosulfates from monoterpenes or sesquiterpenes? Adding some discussions and references regarding this point would be helpful.

4. Page 4 and 11: In this study, did the authors use VBS for SVOC/IVOC species? Did they account for fragmentation and non-volatile SOA? Details regarding this approach need to be added. Also, note that the Cholakian et al. (2017) paper is not available online. Was the treatment of fragmentation and non-volatile SOA similar or different than other papers, for example: [M. Shrivastava et al., 2015; M. Shrivastava et al., 2013].

5. Page 11: What are IVOC/POA emissions assumed in this study? Was oxygen added during aging? How was fragmentation treated? Do their aging mechanisms include NOx-dependence?

6. Page 6: Organic nitrates: How uncertain are the reaction mechanisms and reaction rates for organic nitrate formation? The authors mention organic nitrates were not measured during ChArMex. But, it would be useful to point out what are the most important routes for organic nitrate chemistry mechanism: e.g. relative importance of nighttime TERPNRO2 versus daytime TERPRO2 reaction with NO. Also if this mechanism has been validated against other field campaigns in other papers, those references need to be added.

[Figure]

Interactive
comment

7. Figure 4: What is contribution of SVOC/IVOC to evolution of OM:OC ratio and O:C ratio?

8. Page 14: Discussions on water soluble organics should acknowledge pathways other than organic nitrate to formation of water soluble species. For example, biomass burning sources (if significant), other SOA chemistry e.g. formation of organic salts, aqueous phase and cloud processing etc. See the recent review article on SOA [Shrivastava et al., 2017].

9. Page 20: Hydrolysis is mostly interpreted as a loss mechanism. But Pye et al. 2015 referred to it as pseudo-hydrolysis of organic nitrates that makes them non-volatile. Please clarify if this is what is being referred to.

10. Hydrophylic properties could change not just due to the hydrophobicity of organic nitrates. It could point to other mechanisms and SOA pathways not considered in the model e.g. aqueous and cloud chemistry, other SOA pathways like organic salts, organic-inorganic interactions etc. It is important to make sure the model is getting the right answers for the right reasons.

References: Shrivastava, M., et al. (2017), Recent advances in understanding secondary organic aerosol: Implications for global climate forcing, Rev. Geophys., 55(2), 509-559, doi:10.1002/2016RG000540.

Shrivastava, M., et al. (2015), Global transformation and fate of SOA: Implications of low-volatility SOA and gas-phase fragmentation reactions, J. Geophys. Res.-Atmos., 120(9), 4169-4195, doi:10.1002/2014jd022563.

Shrivastava, M., A. Zelenyuk, D. Imre, R. Easter, J. Beranek, R. A. Zaveri, and J. Fast (2013), Implications of low volatility SOA and gas-phase fragmentation reactions on SOA loadings and their spatial and temporal evolution in the atmosphere, J. Geophys. Res.-Atmos., 118(8), 3328-3342, doi:10.1002/jgrd.50160.

---

## Referee Comment (RC3) · Anonymous Referee #2 · 25 Jul 2017

In this work, Chrit et al. used a mechanistic model to describe the biogenic secondary organic aerosol formation and properties at a measurement supersite in Corsica during two summer campaigns. They found that the consideration of ELVOCs and organic nitrates greatly improved the simulated OA concentrations and oxidation state. This study is of definite interest to the ACP audience by contributing to the organic aerosol modeling field and the scientific methodology used sounds valid. Overall, the manuscript is very well written and the presentation is clear. Therefore, I recommend this study for publication. Below are a few minor comments to be considered prior to publication.

[Figure]

Specific comments:

1. Page 2 line 12: Delete "secondary" before "semi-volatile" and add intermediate volatile organic compounds (IVOCs) as another potential source of SOA.

2. Page 4 line 33: Can you please briefly describe (e.g., using a table) the surrogate species that you are using to represent the SOA formation from these five classes of precursors?

3. Page 5 line 23: It would be interesting to consider this lower yield in your sensitivity analysis.

4. Page 6 Table 1: Are the reported saturation vapor pressures of organic nitrates and MBTCA also at 298 K? If so, please add the "298 K" at the column head.

5. Page 8 line 20: Does this ratio also include the IVOCs? If so, please rewrite as (SVOC+IVOC)/POA

6. Figures 3 and 6: Please consider improving the quality of the figures (e.g., font size, shape of pies, etc.).
* * *

---

## Author Comment (AC1) · 25 Aug 2017

Review of Chrit et al. This paper models the organic aerosol concentration and properties during the ChArMEx summer campaigns of 2012 and 2013 using a surrogate approach for SOA formation. Although I agree with the importance of this topic and its novelty, the modeling approaches need to clearly explained. Also, the caveats of these approaches need to better acknowledged. Below are my specific comments that should be addressed before the paper is accepted for publication.

1. The abstract mentions that surrogate species for monoterpenes are a-pinene and limonene. But on page 5, the authors mention three precursors: a-pinene, b-pinene and limonene. This needs to be clarified. Also, why were these 2 particular species chosen as surrogates for monoterpenes?

Three surrogate species are indeed chosen for monoterpenes: a-pinene, b-pinene and limonene, as detailed in the presentation of the model of this paper and in Pun et al. (2006) and Couvidat et al. (2012). These species are chosen as surrogates for monoterpenes as they are the most abundant monoterpene species emitted, and their abilities to form SOA are different. As the list of monoterpene surrogates is not necessary in the abstract, it is removed, and the sentence "Biogenic precursors are isoprene, monoterpenes (with a-pinene and limonene as surrogate species) and sesquiterpenes." is replaced by "Biogenic precursors are isoprene, monoterpenes and sesquiterpenes."

2. Page 3: Why was IEPOX not observed during ChArMex campaign? Did the instruments measure gas-phase IEPOX or particle phase iepox components? Was the aerosol acidic or neutral?
IEPOX was not measured in the gas phase during the ChArMEx campaign. In Freney et al. (2017), it was estimated in the particle phase using m/z 53 and m/z 82 of AMS measurements. The aerosol is probably acidic over the western Mediterranean region as mentioned by Nicolas José (Thesis), and Couvidat et al. (2013). As postulated in Freney et al. (2017), IEPOX may have reacted to form organosulfates. For clarity, the sentence of the introduction "However, IEPOX was not observed during the ChArMEx campaign over an isoprene-emitting forest in the South of France (Freney et al., 2017)." was replaced by "However, using aerosol mass spectrometer measurements, particle-phase IEPOX was not observed during the ChArMEx campaign over an isoprene-emitting forest in the South of France, suggesting it might have formed organo-sulfates (Freney et al., 2017)."

3. The authors mention the importance of organosulfates in this study. But IEPOX chemistry also makes organosulfates. If these are not coming from isoprene or IEPOX chemistry, what are alternate mechanisms for organosulfates from monoterpenes or sesquiterpenes? Adding some discussions and references regarding this point would be helpful.

Yes, IEPOX may form organo-sulfates. They may also be formed from the oxidation of monoterpenes. A discussion about the formation of organo-sulfates is in section 4.3: "In laboratory, the formation of organosulfate was observed from the uptake of monoterpene

oxidation products (pinonaldehyde) on acidic sulfate aerosols (Liggio and Li, 2006; Surratt et al., 2008), from the uptake of ELVOC (Mutzel et al., 2015), and from the uptake of isoprene oxidation products (Liggio et al., 2005; Nguyen et al., 2014). Isoprene SOA may be formed via the reactive uptake of isoprene-epoxydiol (IEPOX), a second generation oxidation product of isoprene, in the presence of hydrated sulfate (Surratt et al., 2010; Couvidat et al., 2013b; Nguyen et al., 2014). Using aerosol mass spectrometer measurements, Hu et al. (2015) estimated that IEPOX-OA makes a large fraction of the OA (between 17 to 36% in the U.S.) outside urban areas, in agreement with Budisulistiorini et al. (2015). In regions where aerosols are acidic, IEPOX-derived OA may be strongly dependent on the sulfate concentration, which acts as nucleophile and facilitates the ring-opening reaction of IEPOX and organosulfate formation (Nguyen et al., 2014; Xu et al., 2015).
In order to take into account the influence of the formation of organo-sulfates on OA properties, the surrogate products of the model are modified. As Mediterranean aerosol composition displays large concentrations of sulfate, isoprene oxidation products may lead to the formation of organo-sulfate.".

4. Page 4 and 11: In this study, did the authors use VBS for SVOC/IVOC species? Did they account for fragmentation and non-volatile SOA? Details regarding this approach need to be added. Also, note that the Cholakian et al. (2017) paper is not available online. Was the treatment of fragmentation and non-volatile SOA similar or different than other papers, for example: [M. Shrivastava et al., 2015; M. Shrivastava et al., 2013].
In section 2.1 of the revised paper, a brief description of the approach used for SVOC/IVOC species is added. "As detailed in Couvidat et al. (2012), I/S-VOC emissions are emitted in three volatility classes (characterized by their saturation concentrations $c^*$: $\log(c^*)$ = -0.04, 1.93, 3.5). Their ageing is represented through a single oxidation step, without $NO_x$-dependence, to produce species of lower volatilities ($\log(c^*)$ = -2.4, -0.064, 1.5) but higher molecular weights." The VBS approach is not used here, and fragmentation is not taken into account. The paper of Cholakian et al. (2017), which uses a VBS scheme with fragmentation and non-volatile SOA, is about to be published in ACPD.

5. Page 11: What are IVOC/POA emissions assumed in this study? Was oxygen added during aging? How was fragmentation treated? Do their aging mechanisms include NOx-dependence?
IVOCs are considered in our emissions and are included as part of SVOCs. For clarity, the sentences "POA are assumed to be the particle phase of semi-volatile anthropogenic organic emissions (SVOC). Total SVOC emissions are estimated as detailed in Couvidat et al. (2012), by multiplying POA by a fixed value, and by assigning them to species of different volatilities. In this study, the ratio SVOC/POA is set to 2.5 (Kim et al., 2016; Zhu et al., 2016). Setting the ratio SVOC/POA to 1 has little impact on the organic concentrations" are replaced by "POA are assumed to be the particle phase of I/S-VOC. Total I/S-VOC emissions (gas and particle phases) are estimated as detailed in Couvidat et al. (2012), by multiplying POA by a fixed value, and by assigning them to species of different volatilities. In this study, the ratio I/S-VOC/POA is set to 2.5 (Kim et al., 2016; Zhu et al., 2016). Setting the ratio I/S-VOC/POA to 1 has little impact on the organic concentrations, as shown in Figure 2." As now detailed in section 2.1, the ageing of I/S-VOC is represented through a simple approach: a single oxidation step, without $NO_x$-dependence, to produce species of lower volatilities but higher molecular weights (oxygen is added). As there is only one single oxidation step, fragmentation is not considered.

6. Page 6: Organic nitrates: How uncertain are the reaction mechanisms and reaction rates for organic nitrate formation? The authors mention organic nitrates were not measured during ChArMex. But, it would be useful to point out what are the most important routes for organic

nitrate chemistry mechanism: e.g. relative importance of nighttime TERPNRO2 versus daytime TERPRO2 reaction with NO. Also if this mechanism has been validated against other field campaigns in other papers, those references need to be added.

In the organic nitrate formation mechanism developed by Pye et al. (2015), the rate constants are from saprc07 (Carter et al. (2010), Hutzell et al. (2012)) except for the reaction with $HO_2$ which is calculated based on MCM for a species with 10 carbons (Jenkin et al. (1997), Saunders et al. (2003)). The yields of the reactions are taken from SAPRC07 (Carter et al. (2010)). The yields through the TERPNRO2 + NO and $NO_3$ pathways were calculated from estimates of the extent to which the radicals decompose to release $NO_2$ versus retain the nitrate group (Carter et al. (2010)). The yield from the reaction with $HO_2$ produced 100% hydroperoxide (consistent with MCM, (Jenkin et al. (1997), Saunders et al. (2003)). Therefore, it is not easy to determine lower and upper bounds of the rate constants and yields of the reactions. However, as pointed out to in Pye et al., (2015), there is a critical need for additional laboratory data when it comes to the organic nitrate sources (ozonolysis, photooxidation and chemical pathways that do not contain nitrogen). Besides, structure-dependant and pH-dependant hydrolysis rate constants and the resulting product volatility for a range of organic nitrates from biogenic VOCs oxidation.

Moreover, Pye et al. (2015) studied the formation of the organic nitrate within the framework of the Southern Oxidant and Aerosol Study (SOAS) campaign during the summer 2013 over the south-eastern region of the United States (a representative rural site ("Brent", Centreville) in Alabaman using CMAQ model). In Pye et al. (2015), the monoterpene nitrate OA was predicted to be the highest in the south-eastern United States (>95%) and a good agreement was found between the simulated and observed OC and the gas-phase organic nitrates with a bias of -30% and 30% respectively.

The diurnal variation of the organic nitrate is shown in the figure below indicating that the night-time $NO_3$-oxidation might be the most relevant.

[Figure]

The point about the importance of night-time chemistry governing the organic nitrate formation is underlined in section 4.2 of the revised paper by adding:
"The route to organic particulate nitrate may essentially (but not exclusively) be active during the night as $NO_3$ efficiently photolyzes during the day and the production yields are more important during the night, and higher organic-nitrate concentrations are observed at night (Figure 5)."

7. Figure 4: What is contribution of SVOC/IVOC to evolution of OM:OC ratio and O:C ratio? I/S-VOC have very little impact on SOA concentrations and properties in summer. A

sensitivity study with a different I/S-VOC/POA ratio was performed and the Figure below is added to the paper to show the impact of different I/S-VOC emissions on OM concentrations.

[Figure]

Figure 2. Relative difference (%) of OM1 concentrations simulated using the emission ratio I/S-VOC/POA = 2.5 and 1.

8. Page 14: Discussions on water soluble organics should acknowledge pathways other than organic nitrate to formation of water soluble species. For example, biomass burning sources (if significant), other SOA chemistry e.g. formation of organic salts, aqueous phase and cloud processing etc. See the recent review article on SOA [Shrivastava et al., 2017].

Aqueous processing is taken into account in our model, as the affinity with water of most of the surrogates is considered.
To better understand the aqueous treatment of aerosols, a brief description of the gas/particle partitioning of the surrogates is added in section 2.1 of the revised paper. The sentence "For organic aerosols, the partitioning is computed using SOAP (Couvidat and Sartelet, 2015), and bulk equilibrium is also assumed for SOA partitioning." is replaced by "For organic aerosols, the gas/phase partitioning of the surrogates is computed using SOAP (Couvidat and Sartelet, 2015), and bulk equilibrium is also assumed for SOA partitioning. The gas/phase partitioning of hydrophobic surrogates is modelled following Pankow (1994), with absorption by the organic phase (hydrophobic surrogates). The gas/phase partitioning of hydrophylic surrogates is computed using the Henry's law modified to extrapolate infinite dilution conditions to all conditions using an aqueous-phase partitioning coefficient, with absorption by the aqueous phase (hydrophilic organics, inorganics and water). Activity coefficients are computed with the thermodynamic model UNIFAC (UNIversal Functional group, Fredenslund et al., 1975)."

The influence of biomass burning may be low during the periods studied. Organic cloud processing is indeed not considered, although the impact during the summer time over the

Mediterranean domain may be relatively low. Interactions of organics/inorganics are partly considered here (inorganic provide a mass onto which hydrophilic surrogates may condense, but their interactions via activity coefficients are not). In the paper, the discussion shows that the influence of organic nitrate on the hydrophilic properties is high on 16 July compared to 30 July. The hydrophilic properties are indeed influenced by all the surrogates. The following sentences are added in the conclusion to acknowledge pathways that are not considered.

"There are other pathways and mechanisms that are not considered in the model, but that may change the concentrations and hydrophylicity of organics (Shrivastava et al., 2017). For example, salting effects (via activity coefficients) are not considered, although inorganics provide a mass onto which hydrophilic organic surrogates may condense. Furthermore, pathways such as the aging chemistry of VOCs and I/S-VOCs from biomass burning (wildfires) and organic cloud processing are not considered. However, these pathways may be relatively low during the studied periods. "

9. Page 20: Hydrolysis is mostly interpreted as a loss mechanism. But Pye et al. 2015 referred to it as pseudo-hydrolysis of organic nitrates that makes them non-volatile. Please clarify if this is what is being referred to.

Hydrolysis referred here to the pseudo-hydrolysis mechanism detailed in Pye et al. (2015). For clarity, in section 4.4, the sentence "Although this organic nitrate is assumed to be hydrophobic, it may have undergone hydrolysis and formed water soluble compounds." is replaced by "Although this organic nitrate is assumed to be hydrophobic, it may have undergone hydrolysis resulting in nitric acid and nonvolatile secondary organic aerosol that may change the hydrophylicity of organics and the organic composition, as detailed in Pye et al. (2015)."

10. Hydrophylic properties could change not just due to the hydrophobicity of organic nitrates. It could point to other mechanisms and SOA pathways not considered in the model e.g. aqueous and cloud chemistry, other SOA pathways like organic salts, organic-inorganic interactions etc. It is important to make sure the model is getting the right answers for the right reasons.
Other mechanisms and SOA pathways that are not considered in the model, but may change the hydrophylicity of organics are pointed out to in section 4.4 of the revised paper as mentioned in the response of the comment 8.

References:

Carter, W. P. L., Development of the SAPRC-07 chemical mechanism. Atmos. Environ. 2010, 44(40), 5324-5335.

Carter, W. P. L. Development of the SAPRC-07 Chemical Mechanism and Updated Ozone Reactivity Scales, Final report to the California Air Resources Board Contract No. 03−318. January 27, 2010,www.cert.ucr.edu/~carter/SAPRC.

Cholakian, A., Beekmann, M., Colette, A., Coll, I., Siour, G., Sciare, J., Marchand, N., Pey, J., Gros, V., Sauvage, S., Sellegri, K., Colomb, A., Sartelet, K., and Dulac, F.: Simulation of organic aerosols in the western Mediterranean area during the ChArMEx 2013 summer campaign, Atmos. Chem. Phys. Discuss., p. submitted, 2017

Couvidat, F., Sartelet, K., and Seigneur, C.: Investigating the impact of aqueous-phase chemistry and wet deposition on organic aerosol formation using a molecular surrogate modeling approach, Environ. Sci. Technol., 47, 914–922, doi:10.1021/es3034318, 2013.

Grieshop, A. P., N. M. Donahue, and A. L. Robinson (2009a), Laboratory investigation of photochemical oxidation of organic aerosol from wood fires 2: Analysis of aerosol mass spectrometer data, Atmos. Chem. Phys., 9(6), 2227–2240, doi:10.5194/acp-9-2227-2009.

Hutzell, W. T.; Luecken, D. J.; Appel, K. W.; Carter, W. P. L., Interpreting predictions from the SAPRC07 mechanism based on regional and continental simulations. Atmos. Environ. 2012, 46 (0), 417- 429.

Jenkin, M. E.; Saunders, S. M.; Pilling, M. J., The tropospheric degradation of volatile organic compounds: A protocol for mechanism development. Atmos. Environ. 1997, 31 (1), 81-104.

Pun, B., C. Seigneur, and K. Lohman, Modeling secondary organic aerosol formation via multiphase partitioning with molecular data, Environ. Sci. Technol., 40, 4722–4731, doi:10.1021/es0522736, 2006.

Pye, H., Luecken, D. J., Xu, L., Boyd, C., Ng, N., Baker, K., Ayres, B., Bash, J., Baumann, K., Carter, W., Edgerton, E., Fry, J., Hutzell, W., 10 Schwede, D., and Shepson, P.: Modelling the current and the future roles of particulate organic nitrates in the southeastern United States, Environ. Sci. Technol., 49, 14 195–14 203, doi:10.1021/acs.est.5b03738, 2015.

Robinson, A. L., Donahue, N. M., Shrivastava, M. K., Weitkamp, E. A., Sage, A. M., Grieshop, A. P., Lane, T. E., Pierce, J. R., and Pandis, S. N.: Rethinking organic aerosols: semivolatile emissions and photochemical aging, Science, 315, 1259–1262, doi:10.1126/science.1133061, 2007.

Saunders, S. M.; Jenkin, M. E.; Derwent, R. G.; Pilling, M. J., Protocol for the development of  the Master Chemical Mechanism, MCM v3 (Part A): tropospheric degradation of non-aromatic volatile organic compounds. Atmos. Chem. Phys. 2003, 3, 161-180.

Shrivastava, M., et al. (2017), Recent advances in understanding secondary organic aerosol: Implications for global climate forcing, Rev. Geophys., 55(2), 509-559, doi:10.1002/2016RG000540.

Shrivastava, M., et al. (2015), Global transformation and fate of SOA: Implications of low-volatility SOA and gas-phase fragmentation reactions, J. Geophys. Res.-Atmos., 120(9), 4169-4195, doi:10.1002/2014jd022563.

Shrivastava, M., A. Zelenyuk, D. Imre, R. Easter, J. Beranek, R. A. Zaveri, and J. Fast (2013), Implications of low volatility SOA and gas-phase fragmentation reactions on SOA loadings and their spatial and temporal evolution in the atmosphere, J. Geophys. Res.-Atmos., 118(8), 3328-3342, doi:10.1002/jgrd.50160.

---

## Author Comment (AC2) · 25 Aug 2017

In this work, Chrit et al. used a mechanistic model to describe the biogenic secondary organic aerosol formation and properties at a measurement supersite in Corsica during two summer campaigns. They found that the consideration of ELVOCs and organic nitrates greatly improved the simulated OA concentrations and oxidation state. This study is of definite interest to the ACP audience by contributing to the organic aerosol modeling field and the scientific methodology used sounds valid. Overall, the manuscript is very well written and the presentation is clear. Therefore, I recommend this study for publication. Below are a few minor comments to be considered prior to publication.

Specific comments:

1. Page 2 line 12: Delete "secondary" before "semi-volatile" and add intermediate volatile organic compounds (IVOCs) as another potential source of SOA.
The sentences "OA are classified either as primary (POA) or as secondary aerosols (SOA). POA are directly emitted in the atmosphere, whereas SOA are produced through chemical oxidation of volatile organic compounds (VOCs) and secondary semi-volatile organic compounds (SVOCs). SOA are often semi volatile, i.e. they partition between the gas and particle phases." are replaced by "OA are usually classified either as primary (POA) or as secondary aerosols (SOA). POA are directly emitted in the atmosphere, often as intermediate/semi-volatile organic compounds (I/S-VOCs), which partition between the gas and the particle phases (Robinson et al., 2007). The gas-phase I/S-VOC are missing from emission inventories (Couvidat et al. 2012, Kim et al. 2016). SOA are produced through chemical oxidation of volatile organic compounds (VOCs) and I/S-VOCs, and condensation of I/S-VOCs.

2. Page 4 line 33: Can you please briefly describe (e.g., using a table) the surrogate species that you are using to represent the SOA formation from these five classes of precursors?
A table of the five classes of precursors is added in the revised paper:

| Precursors | Surrogate species |
|---|---|
| I/S-VOC | 3 volatility bins : log(c*) = -0.04, 1.93, 3.5 with c* the saturation concentration. |
| Aromatics | Toluene, xylene |
| Isoprene | Isoprene |
| Monoterpenes | α-pinene, β-pinene, limonene |
| Sesquiterpenes | Humulene |

3. Page 5 line 23: It would be interesting to consider this lower yield in your sensitivity analysis.

It is possible to infer a lower and upper bound from the papers of Ehn et al. (2014) et Jokinen et al. (2015), as shown in the Table below.

| VOCs | Ehn et al. (2014) | Jokinen et al. (2015) |
|---|---|---|
| α-pinene | 7% ± 3.5% | 3.4% ± 1.7% |
| Limonene | 17% ± 8.5% | 5.3% ± 2.6% |

Details on the ELVOC yield and the choice of the bounds are added in section 2.2. The sentence "ELVOC is assumed to be formed with an average molar yield of 11% following Ehn et al. (2014), although Jokinen et al. (2015) reported lower yields (about 5%)." is replaced by "The ELVOC yield is assumed to be 11%, i.e. close to the average of the yields of a-pinene and limonene according to Ehn et al. (2014). Jokinen et al. (2015) suggested lower yields (Table 2). In this paper, sensitivity simulations with a lower bound of 3% and a upper bound of 18% are also conducted".

The following sentences are added in section 4. "Two sensitivity simulations are performed using a lower bound yield (3%) and an upper bound yield (18%). In Appendix D, similarly to what is presented in this section for the reference simulation, the sensitivity simulations are compared to each other and to the measurements in terms of the mass of $OM_1$, the organic aerosol composition, the OM:OC and O:C ratios.

4. Page 6 Table 1: Are the reported saturation vapor pressures of organic nitrates and MBTCA also at 298 K? If so, please add the "298 K" at the column head.
The "298 K" is added for the reported saturation vapor pressure of MBTCA in table 1 of the revised paper.

5. Page 8 line 20: Does this ratio also include the IVOCs? If so, please rewrite as (SVOC+IVOC)/POA
For clarity, the sentences "POA are assumed to be the particle phase of semi-volatile anthropogenic organic emissions (SVOC). Total SVOC emissions are estimated as detailed in Couvidat et al. (2012), by multiplying POA by a fixed value, and by assigning them to species of different volatilities. In this study, the ratio SVOC/POA is set to 2.5 (Kim et al., 2016; Zhu et al., 2016). Setting the ratio SVOC/POA to 1 has little impact on the organic concentrations" are replaced by "POA are assumed to be the particle phase of I/S-VOC. Total I/S-VOC emissions (gas and particle phases) are estimated as detailed in Couvidat et al. (2012), by multiplying POA by a fixed value, and by assigning them to species of different volatilities. In this study, the ratio I/S-VOC/POA is set to 2.5 (Kim et al., 2016; Zhu et al., 2016). Setting the ratio I/S-VOC/POA to 1 has little impact on the organic concentrations, as shown in Figure 5."

6. Figures 3 and 6: Please consider improving the quality of the figures (e.g., font size, shape of pies, etc.).
The quality of the figures is improved in the revised paper.

References:

Couvidat, F., Debry, É., Sartelet, K., and Seigneur, C.: A hydrophilic/hydrophobic organic (H2O) model: Model development, evaluation and sensitivity analysis, J. Geophys. Res., 117, D10 304, doi:10.1029/2011JD017214, 2012

Ehn, M., Thornton, J., Kleist, E., Sipilä, M., Junninen, H., Pullinen, I., Springer, M., Rubach, F., Tillmann, R., Lee, B., Lopez-Hilfiker, F., Andres, S., Acir, I., Rissanen, M., Jokinen, T., Schobesberger, S., Kangasluoma, J., Kontkanen, J., Nieminen, T., Kurtén, T., Nielsen, 10 L. B., Jørgensen, S., Kjaergaard, H. G., Canagaratna, M., Dal Maso, M., Berndt, T., Petäjä, T., Wahner, A., Kerminen, V., Kulmala, M., Worsnop, D. R., Wildt, J., and Mentel, T. F.: A large source of low-volatility secondary organic aerosol, Nature, 506, 476–479, doi:10.1038/nature13032, 2014.

Jokinen, T., Berndt, T., Makkonen, R., Kerminen, V., Junninen, H., Paasonen, P., Stratmann, F., Herrmann, H., Guenther, A. B., Worsnop, D. R., Kulmala, M., Ehn, M., and Sipiläb, M.: Production of extremely low volatile organic compounds from biogenic emissions: Measured yields and atmospheric implications, Proc. Nat. Acad. Sci., 112, 7123–7128, doi:10.1073/pnas.1423977112, 2015.

---

## Author Comment (AC3) · 25 Aug 2017

Chrit et al. have performed simulations using an air quality model and compared predictions of organic aerosol mass and composition to measurements at a remote site in the Mediterranean Sea. They found that model updates based on the inclusion of new secondary organic aerosol formation pathways improved the model-measurement comparison. Overall the manuscript is well motivated, researched, and discussed. I recommend publication after the authors have had an opportunity to respond to my comments and considered my suggestions.

1. Page 1, line 5: Consider using the noun form: hydrophilicity
"oxidation state and hydrophilic" is replaced in the revised version by "oxidation state and hydrophilicity"

2. Page 1, line 15: 'percent' not 'percents'.
"Percents" is replaced in the revised paper by "Percent".

3. Page 2, lines 1-10: References are dated. Consider newer references.
Jimenez et al., 2009, Lin et al., 2014 and Tuet et al., 2017 are added in the revised version.

4. Page 2, lines 8-9: Rephrase.
In the revised version, we replaced the sentence "Considering health effects, oxidative stress, which is induced by the generation of reactive oxygen species (ROS), is suggested as one pathway of OA toxicity." by "In terms of health effects, OA toxicity is linked to the oxidative stress which is induced by the ROS (Reactive Oxygen Species)".

5. Page 2, lines 13: Intermediate-volatility organic compounds are a separate precursor category. Add discussion in the introduction. A re
The sentences "OA are classified either as primary (POA) or as secondary aerosols (SOA). POA are directly emitted in the atmosphere, whereas SOA are produced through chemical oxidation of volatile organic compounds (VOCs) and secondary semi-volatile organic compounds (SVOCs). SOA are often semi volatile, i.e. they partition between the gas and particle phases." are replaced by "OA are usually classified either as primary (POA) or as secondary aerosols (SOA). POA are directly emitted in the atmosphere, often as intermediate/semi-volatile organic compounds (I/S-VOCs), which partition between the gas and the particle phases (Robinson et al., 2007). The gas-phase I/S-VOC are missing from emission inventories (Couvidat et al. 2012, Kim et al. 2016). SOA are produced through chemical oxidation of volatile organic compounds (VOCs) and I/S-VOCs, and condensation of I/S-VOCs.

6. Page 2, lines 16-22: It would be helpful to be more quantitative when citing earlier work. For example, what fraction of the organic aerosol that El Haddad et al. (2011, 2013) measured was biogenic in nature?
The sentence "El Haddad et al. (2011, 2013) attributed most of the organic carbon (OC) mass to biogenic secondary organic carbon (BSOC), and to monoterpene oxidation products." is replaced by "El Haddad et al. (2011, 2013) attributed 80% of the organic aerosol mass to biogenic secondary organic aerosols (BSOA), and they attributed near 40% of the BSOA to monoterpene oxidation products."

7. Page 2, lines 17-19: Hayes et al. (2013, 2015) argue that the biogenic SOA found in Los Angeles was produced near the source and then transported into the city. Suggest citing and reconciling with Hayes work.

A reference to the work of Hayes et al. (2015) is added in the revised paper. Hayes et al. (2013, 2015) characterized the organic aerosol composition and sources in Pasadena in California during the 2010 CalNex campaign and found a substantial contribution from regional biogenic SOA. Biogenic emissions are transported and age during transport, for example during transport over the Central Valley (where there are anthropogenic emissions) in the case of Pasadena.

A sentence is added to the manuscript as followed to cite the work of Hayes et al. (2015) "A large fraction of emitted VOCs is biogenic, especially in the western Mediterranean in summer, when solar radiation is high. Biogenic emissions may age and form SOA as they are transported through different environments (Hayes et al., 2015)."

8. Page 2, line 20, Page 9, line23 (and elsewhere too): 'Fossil' and not 'fossile'.

"Fossile" is replaced by "fossil" throughout the revised paper.

9. Page 2, line 19: Was Minguillon a measurement or modeling study?

Minguillon et al. (2016) is an experimental study carried out in Barcelona in summer 2013. For clarity, the sentence "Similar results were obtained in a campaign in the Barcelona region (Spain), where Minguillón et al. (2011) have found a prevalence of non-fossil organic aerosol sources in remote and urban environments, and a clear evidence of biogenic VOC oxidation products and biogenic SOA formation under anthropogenic stressors (Minguillón et al., 2016)." is replaced by "Similar results were obtained through measurement campaigns in the Barcelona region (Spain), where Minguillón et al. (2011, 2016) have found a prevalence of non-fossil organic aerosol sources in remote and urban environments, and a clear evidence of biogenic VOC oxidation products and biogenic SOA formation under anthropogenic stressors."

10. Page 3, line 16: Incomplete sentence: 'monoterpenes oxidation products SOA over the U.S.'.

In the revised version, "organic nitrate accounts for more than a half of the monoterpene oxidation products SOA over the U.S." is replaced by "organic nitrate accounts for more than a half of the monoterpene oxidation products in the particle phase over the U.S."

11. Page 4, Section 2.1: Can you briefly summarize the existing SOA precursors, species, and processes in the model and discuss how those earlier processes do not overlap with the updates made in this work?

In section 2.1 of the revised paper, a brief description of the existing SOA anthropogenic precursors, their aging processes and products is added.

"As detailed in Couvidat et al. (2012), I/S-VOC emissions are emitted in three volatility classes (characterized by their saturation concentrations $c^*$: $\log(c^*)$ = -0.04, 1.93, 3.5). Their ageing is represented through a single oxidation step, without $NO_x$-dependence, to produce species of lower volatilities ($\log(c^*)$ = -2.4, -0.064, 1.5) but higher molecular weights. For aromatic compounds, toluene and xylene are used as surrogate precursors. The precursors react with OH to form radicals that may then react differently under low-NOx and high-NOx conditions. Under low-NOx conditions, the surrogate is not identified, but it is supposed to be hydrophobic. Under high-$NO_x$ conditions, the surrogate formed are two benzoic acids (methyl nitro benzoic acid and methyl hydroxyl benzoic acid).

12. Page 5, line 26-28: Am I correct that these oxygenated peroxy radicals can be formed only in the absence of NO? If yes, specify.

Peroxy radicals $RO_2$ are preferentially formed in low-$NO_x$ conditions, where O3 concentration is high. However, depending on the NOx concentrations, they may still form in high-NOx conditions, and then react with NO to form organic nitrate (reaction (A9) of Appendix A). Organic nitrate from ELVOCs were not considered here because their concentrations were negligible.

13. How is gas/particle partitioning of the explicit oxidation products modeled? If only briefly, please summarize the partitioning model and assumptions.

A brief description of the gas/particle partitioning of the surrogates is added in section 2.1 of the revised paper. The sentence "For organic aerosols, the partitioning is computed using SOAP (Couvidat and Sartelet, 2015), and bulk equilibrium is also assumed for SOA partitioning." is replaced by "For organic aerosols, the gas/phase partitioning of the surrogates is computed using SOAP (Couvidat and Sartelet, 2015), and bulk equilibrium is also assumed for SOA partitioning. The gas/phase partitioning of hydrophobic surrogates is modelled following Pankow (1994), with absorption by the organic phase (hydrophobic surrogates). The gas/phase partitioning of hydrophylic surrogates is computed using the Henry's law modified to extrapolate infinite dilution conditions to all conditions using an aqueous-phase partitioning coefficient, with absorption by the aqueous phase (hydrophilic organics, inorganics and water). Activity coefficients are computed with the thermodynamic model UNIFAC (UNIversal Functional group,  Fredenslund et al., 1975)."

14. Page 8, line 2: I did not understand the meaning of 'nested rep.' in parentheses.

In the revised version "(nested rep.)" is replaced by "nested respectively" and "(6 June and 8 July 2012 resp.)" by "(6 June and 8 July 2012 respectively)".

15. Page 8, line 14: 'split' not 'splitted'.

"splitted" is replaced by "split" in the revised version.

16. Page 8, line 18-20: The SVOC/POA ratios might be a little too high. See work of May et al. (2014a,b) for estimates on SVOC/POA ratios. Also, is there a reason why IVOC emissions were not considered? See work of Jathar et al. (2014) and Zhao et al. (2015, 2106) for estimates of IVOCs from combustion sources as a function of VOC emissions.

IVOCs are considered in our emissions and are included as part of SVOCs. Moreover, in this paper, the I/S-VOC/POA ratio corresponds to the ratio of (gas+ particle phases)/(particle phase). This ratio equals to 1.5 if only the gas-phase of I/S-VOC is considered in the ratio I/S-VOC/POA. The ratio equals 2.5 if both gas and particle phases of I/S-VOC are considered in the ratio I/S-VOC/POA. The choice made is based on the study of Kim et al 2006 who evaluated experimentally the ratio for exhaust emissions from gasoline and diesel vehicles, and from the air-quality simulations of Zhu et al. (2016) over Greater Paris.

For clarity, the sentences "POA are assumed to be the particle phase of semi-volatile anthropogenic organic emissions (SVOC). Total SVOC emissions are estimated as detailed in Couvidat et al. (2012), by multiplying POA by a fixed value, and by assigning them to species of different volatilities. In this study, the ratio SVOC/POA is set to 2.5 (Kim et al., 2016; Zhu et al., 2016). Setting the ratio SVOC/POA to 1 has little impact on the organic concentrations" are replaced by "POA are assumed to be the particle phase of I/S-VOC. Total I/S-VOC emissions (gas and particle phases) are estimated as detailed in Couvidat et al. (2012), by multiplying POA by a fixed value, and by assigning them to species of different volatilities. In this study, the ratio I/S-VOC/POA is set to 2.5 (Kim et al., 2016; Zhu et al., 2016). Setting the ratio I/S-VOC/POA to 1 has little impact on the organic concentrations, as shown in Figure 2."

17. Page 8, lines 25-28: What is the organic fraction in sea salt emissions?

The sentence "The organic fraction of sea-salt emissions is not taken into account in the simulation presented here. However, it is estimated in section 5, where the contribution of organic sea-salt emissions to organic concentrations is assessed." is added after line 28 of page 8 to the revised version of the paper.

18. Page 10: How does the model perform in predicting the diurnal variations in OA?

The figure below illustrates the model-to-measurement comparison of the diurnal concentrations of $OM_1$ concentrations. There is no clear diurnal variation in the measurements, probably because the organic compounds are very oxidized and of low volatility. The model slightly underestimates the

[Figure]

Diurnal variations in OM$_1$ concentrations ($\mu$g.m$^{-3}$) in 2013

19. Figure 2: The composition in Figure 3 tells me that the model updates must also have increased mass concentrations and resulted in better comparison in Figure 2. This fact is missing in Section 4.1. Please state the importance of this either in Section 4.1 or in the conclusions in Section 6.
This remark is added in section 6 (conclusion) of the revised paper. "During the summer 2013, the added surrogates contribute to 15% of the OM$_1$ mass for ELVOC, 20% for organic nitrate from monoterpene oxydation and 0.2% for MBTCA. In agreement with $^{14}$C measurements, most of the organic aerosol is from non-fossil (biogenic) origin."

20. Section 4.2: Has the model been evaluated for other pollutants? For example, black carbon, carbon monoxide, ozone, NO.
The model was also evaluated for other pollutants, like the concentrations of other particle components and precursors, and ozone. The ozone is slightly overestimated (the measured and simulated means are 51.30 $\mu$g/m$^3$ and 60.69 $\mu$g/m$^3$ respectively). The black carbon in PM$_{2.5}$ is also evaluated and found to be overestimated (the measured and simulated means are 0.28 $\mu$g/m$^3$ and 0.56 $\mu$g/m$^3$ respectively).
Another paper about the origins of particles and comparisons to measurements of the concentrations of particle components and precursors is about to be submitted. Therefore, these comparisons are not added to this paper.

21. Page 11, line 20: How low are the sesquiterpene emissions compared to isoprene and monoterpenes? Be quantitative.

In section 3.1.3 of the revised paper, the following sentence is added "… from Nature (MEGAN, Guenther et al (2006)). Over, the Mediterranean domain, during the period of the 2013 summer simulation, the mean emissions of sesquiterpenes, monoterpenes and isoprene are 0.001, 0.019 and 0.024 $\mu$g/m$^2$/s respectively. Hence, comparing to isoprene and. monoterpene emissions, the sesquiterpene emissions are lower by a factor of 95.8% and 94.7% respectively."

22. Page 11, line 22: IVOCs are mentioned here but are they actually included since the methods do not talk about them.
As detailed in the previous replies, in the ACPD version of the paper, SVOC emissions referred to both I/S-SVOC. Details on the modelling are added: "As detailed in Couvidat et al. (2012), I/S-VOC emissions are emitted in three volatility classes (characterized by their saturation concentrations c*: log(c*) = -0.04, 1.93, 3.5). Their ageing is represented through a single oxidation

23. Figure 3 and 6, left panel: Why is the pie not a circle? Also, consider increasing the font size for readability.
The setting to include the figures in the paper is modified, so that the pie looks like a circle. The font-size in figures 3 and 6 is increased for readability sake.

24. Page 12, line 5: 'There are a variety. . .'.
"There are variety …" is modified to "There is a variety …"

25. Page 12, line 7: 'ponderating'?
"… ponderating …" is replaced in the revised paper by "… weighting …"

26. Figure 4: The legend makes it seem like the model predictions systematically discount (or subtract) the effects of the three updates rather than the opposite that is mentioned in the caption. Replace '-' with '+'?
In the legend of figure 4 of the revised paper, the "-" are replaced by "+".

27. Figure 4: Given that the OM:OC and O:C measurements are not directly measured but rather interpreted from the ACSM data, can the measurements be shown with error bars?
As stated by Crenn et al. (2015) from the intercomparison of 13 Q-ACSMs (including the one used in this study), an important instrument-to-instrument variability is observed in the O/C ratio. This variability currently remains unexplained; it appears to be independent of the organic mass concentrations and could be due to instrument-dependent differences in the vaporization conditions. Because of this instrument-to-instrument variability, and because it is difficult to define a reference instrument which can provide an accurate measurement of the O/C ratio, a correct estimate of our Q-ACSM measurement uncertainty of the O/C ratio remains hard to provide.
However, and further to the discussion provided in the manuscript, an indicative OM/OC ratio can be calculated here from the direct comparison of OM (from Q-ACSM) and OC (from OCEC Sunset field instrument). Comparison was performed during the campaign, for a 3-week period (15/07-05/08/2013), and showed a slope of 2.0 (r²=0.80; N = 252 valid data points).

[Figure]

 OC measurements were obtained at PM2.5 instead of PM1 for OM (Q-ACSM). This may lead to higher OM/OC ratio (in PM1). Based on co-located OC size-segregated measurements performed during the campaign by low-pressure 13-stage DEKATI cascade impactor, we have calculated that OC in PM2.5 was typically 10% higher compared to OC in PM1 (J. Sciare, personal communication). This would result in an experimentally determined OM/OC ratio of 2.2, which

28. Page 14, lines 5-7: Can the authors describe the mechanism at play here and the correct citation?
A brief description of the gas/particle partitioning of the surrogates is added in section 2.1 of the revised paper: "For organic aerosols, the gas/phase partitioning of the surrogates is computed using SOAP (Couvidat and Sartelet, 2015), and bulk equilibrium is also assumed for SOA partitioning. The gas/phase partitioning of hydrophobic surrogates is modelled following Pankow (1994), with absorption by the organic phase (hydrophobic surrogates). The gas/phase partitioning of hydrophylic surrogates is computed using the Henry's law modified to extrapolate infinite dilution conditions to all conditions using an aqueous-phase partitioning coefficient, with absorption by the aqueous phase (hydrophilic organics, inorganics and water). Activity coefficients are computed with the thermodynamic model UNIFAC (UNIversal Functional group, Fredenslund et al., 1975). Therefore, the concentrations in the aqueous phase increase when the concentrations of inorganics (particularly sulfate) increase.

In section 4.3 of the revised paper, the sentence "Furthermore, a large part of biogenic SOA is hydrophilic and therefore higher condensation of sulfate enhances their partitioning into the particulate phase." is replaced by "Furthermore, a large part of biogenic SOA is hydrophilic and therefore higher condensation of sulfate enhances their partitioning into the particulate phase, as the mass of the aqueous phase increases through the condensation of sulfate (Couvidat and Sartelet (2015)."

29. Section 4.4 and Page 1, line 11: The claims about improving the hydrophilicity predictions are slightly misleading since what the authors have actually done is improve predictions of mass concentrations of water soluble organic carbon.
Yes, indeed. The improvement concerns the mass concentrations of water-soluble organic carbon. "… oxidation state and hydrophilic properties …" is replaced by "… oxidation property of organics and the hydrophilic organic carbon…" in the revised version of the paper.

30. Figure 6 does not have a left-right panel but a top-bottom panel. Fix caption.
The panels in Figure 6 are modified to not be top-bottom but left-right.

31. Figures 7, 8 and 9 could be combined into a single multi-panel figure. Also, consider adding city names to orient the reader not familiar with that part of the world.
Figures 7, 8, and 9 are combined into a multi-panel figure. Besides, the island name (Corsica), the Mediterranean sea, the Adriatic sea, the name of FRANCE and ITALY are added in order to orient the reader who are not familiar with that part of the world.

32. Sensitivity analysis: How sensitive are the model predictions and the findings from this work to the various inputs (reaction rate constants, yields, etc.) listed in the appendix? I would encourage the authors to perform additional simulations to (i) develop lower and upper bounds on their estimates for ELVOCs, organic nitrates, and MBTCA and (ii) develop insight on the most important inputs that would guide future laboratory work.

There are uncertainties in the reactions rate constants and yields of the mechanisms leading to the formation of ELVOCs, organic nitrate and MBTCA. However, those uncertainties are difficult to evaluate.

For the formation of ELVOCs and the autoxidation mechanism added in the model, the reaction rate constants are defined in Ehn et al. (2014). In fact, $k_1$ of the ozonolysis reaction is suggested by MCM, $k_{H/O2}$ is calculated in Ehn et al. (2014) to reproduce the turnover at the same point as in the observations during the chamber experiments. It is not straightforward to determine lower and upper bounds of these reaction rate constants. Concerning the yields of ELVOCs, it is possible to infer a lower and upper bound from the papers of Ehn et al. (2014) and Jokinen et al. (2015), as

| VOCs | Ehn et al. (2014) | Jokinen et al. (2015) |
|------|-------------------|------------------------|
| α-pinene | 7% ± 3.5% | 3.4% ± 1.7% |
| Limonene | 17% ± 8.5% | 5.3% ± 2.6% |

Details on the ELVOC yield and the choice of the bounds are added in section 2.2. The sentence "ELVOC is assumed to be formed with an average molar yield of 11% following Ehn et al. (2014), although Jokinen et al. (2015) reported lower yields (about 5%)." is replaced by "The ELVOC yield is assumed to be 11%, i.e. close to the average of the yields of a-pinene and limonene according to Ehn et al. (2014). Jokinen et al. (2015) suggested lower yields (Table 2). In this paper, sensitivity simulations with a lower bound of 3% and a upper bound of 18% are also conducted".

The following sentences are added in section 4. "Two sensitivity simulations are performed using a lower bound yield (3%) and an upper bound yield (18%). In Appendix D, similarly to what is presented in this section for the reference simulation, the sensitivity simulations are compared to each other and to the measurements in terms of the mass of $OM_1$, the organic aerosol composition, the OM:OC and O:C ratios.

In the organic nitrate formation mechanism developed by Pye et al. (2015), the rate constants are from saprc07 (Carter et al. (2010), Hutzell et al. (2012)) except for the reaction with $HO_2$ which is calculated based on MCM for a species with 10 carbons (Jenkin et al. (1997), Saunders et al. (2003)). The yields of the reactions are taken from SAPRC07 (Carter et al. (2010)). The yields through the TERPNRO2 + NO and $NO_3$ pathways were calculated from estimates of the extent to which the radicals decompose to release $NO_2$ versus retain the nitrate group (Carter et al. (2010)). The yield from the reaction with $HO_2$ produced 100% hydroperoxide (consistent with MCM, (Jenkin et al. (1997), Saunders et al. (2003)). Therefore, it is not easy to determine lower and upper bounds of the rate constants and yields of the reactions. However, as pointed out to in Pye et al., (2015), there is a critical need for additional laboratory data when it comes to the organic nitrate sources (ozonolysis, photooxidation and chemical pathways that do not contain nitrogen). Besides, structure-dependant and pH-dependant hydrolysis rate constants and the resulting product volatility for a range of organic nitrates from biogenic VOCs oxidation.

The formation of MBTCA is very low here. Therefore, as stated in the conclusion, further sensitivity studies should not focus on determining a single yield for MBTCA, as done here, but a yield for carboxylic acids that may partition to the particle phase.

References:

Carter, W. P. L., Development of the SAPRC-07 chemical mechanism. Atmos. Environ. 2010, 44(40), 5324-5335.

Carter, W. P. L. Development of the SAPRC-07 Chemical Mechanism and Updated Ozone Reactivity Scales, Final report to the California Air Resources Board Contract No. 03−318. January 27, 2010,www.cert.ucr.edu/~carter/SAPRC.

Couvidat, F., Debry, É., Sartelet, K., and Seigneur, C.: A hydrophilic/hydrophobic organic (H2O) model: Model development, evaluation and sensitivity analysis, J. Geophys. Res., 117, D10 304, doi:10.1029/2011JD017214, 2012

F. Couvidat and K. Sartelet, The Secondary Organic Aerosol Processor (SOAP v1.0) model: a unified model with different ranges of complexity based on the molecular surrogate approach, Geosci. Model Dev., 8, 1111−1138, 2015, doi:10.5194/gmd-8-1111-2015

V. Crenn , J. Sciare , P. L. Croteau , S. Verlhac , R. Fröhlich , C. A. Belis , W. Aas , M. Äijälä, A.

Herrmann, C. Lunder, M. C. Minguillón, G. Mocnik , C. D. O'Dowd, J. Ovadnevaite, J.-E. Petit1, E. Petralia, L. Poulain, M. Priestman, V. Riffault, A. Ripoll, R. Sarda-Estève, J. G. Slowik, A. Setyan, A. Wiedensohler, U. Baltensperger, A. S. H. Prévôt, J. T. Jayne, and O. Favez: ACTRIS ACSM intercomparison − Part 1: Reproducibility of concentration and fragment results from 13 individual Quadrupole Aerosol Chemical Speciation Monitors (Q-ACSM) and consistency with co-located instruments, Atmos. Meas. Tech., 8, 5063–5087, doi:10.5194/amt-8-5063-2015, 2015.

Ehn, M., Thornton, J., Kleist, E., Sipilä, M., Junninen, H., Pullinen, I., Springer, M., Rubach, F., Tillmann, R., Lee, B., Lopez-Hilfiker, F., Andres, S., Acir, I., Rissanen, M., Jokinen, T., Schobesberger, S., Kangasluoma, J., Kontkanen, J., Nieminen, T., Kurtén, T., Nielsen, 10 L. B., Jørgensen, S., Kjaergaard, H. G., Canagaratna, M., Dal Maso, M., Berndt, T., Petäjä, T., Wahner, A., Kerminen, V., Kulmala, M., Worsnop, D. R., Wildt, J., and Mentel, T. F.: A large source of low-volatility secondary organic aerosol, Nature, 506, 476–479, doi:10.1038/nature13032, 2014.

Fredenslund, Aa., Jones, R. L. and Prausnitz, J. M., Group-Contribution Estimation of Activity Coefficients in Nonideal Liquid Mixtures. AIChE J., 21, 1086-1099 (1975).

Jenkin, M. E.; Saunders, S. M.; Pilling, M. J., The tropospheric degradation of volatile organic compounds: A protocol for mechanism development. Atmos. Environ. 1997, 31 (1), 81-104.

Jokinen, T., Berndt, T., Makkonen, R., Kerminen, V., Junninen, H., Paasonen, P., Stratmann, F., Herrmann, H., Guenther, A. B., Worsnop, D. R., Kulmala, M., Ehn, M., and Sipiläb, M.: Production of extremely low volatile organic compounds from biogenic emissions: Measured yields and atmospheric implications, Proc. Nat. Acad. Sci., 112, 7123–7128, doi:10.1073/pnas.1423977112, 2015.

Hutzell, W. T.; Luecken, D. J.; Appel, K. W.; Carter, W. P. L., Interpreting predictions from the SAPRC07 mechanism based on regional and continental simulations. Atmos. Environ. 2012, 46 (0), 417- 429.

Nieminen, T., Kurtén, T., Nielsen, 10 L. B., Jørgensen, S., Kjaergaard, H. G., Canagaratna, M., Dal Maso, M., Berndt, T., Petäjä, T., Wahner, A., Kerminen, V., Kulmala, M., Worsnop, D. R., Wildt, J., and Mentel, T. F.: A large source of low-volatility secondary organic aerosol, Nature, 506, 476–479, doi:10.1038/nature13032, 2014.

Pye, H., Luecken, D. J., Xu, L., Boyd, C., Ng, N., Baker, K., Ayres, B., Bash, J., Baumann, K., Carter, W., Edgerton, E., Fry, J., Hutzell, W., 10 Schwede, D., and Shepson, P.: Modelling the current and the future roles of particulate organic nitrates in the southeastern United States, Environ. Sci. Technol., 49, 14 195–14 203, doi:10.1021/acs.est.5b03738, 2015.

Saunders, S. M.; Jenkin, M. E.; Derwent, R. G.; Pilling, M. J., Protocol for the development of the Master Chemical Mechanism, MCM v3 (Part A): tropospheric degradation of non-aromatic volatile organic compounds. Atmos. Chem. Phys. 2003, 3, 161-180.

---

## Author Response (AR2)

1. Some of the responses on the pdf at the bottom of the page are cut off. Also some of the citations in the response do not show up in the bibliography.
The pdf file of the response to reviewer 3 was rebuilt to correctly show all pages. Not all the citations used in the response to reviewer are used in the article. Only the citations that are used in the article are shown in the bibliography of the article. However, at the end of each response to reviewer, we added the citations that are discussed in the response to reviewer.

2. Comment 7: I do not think Hayes, 2015 states a high fraction of organic aerosol attributed to biogenic SOA in Pasadena. Please confirm.
Indeed, Hayes et al. 2015 found that local biogenic VOCs are predicted to contribute only a few percent to SOA in Pasadena. However, a regional SOA background (of approximately 2.1 $\mu g\ m^{-3}$) is also present due to the long-distance transport of highly aged OA, likely with a substantial contribution from regional biogenic SOA.

3. Comment 16: While the authors have changed the text 'SVOC' to 'I/S-VOC' everywhere in the manuscript, they still don't account for IVOCs explicitly since the two papers they refer to (Kim, 2016 and Zhu, 2016) do not include any primary references to IVOC measurements. I would like the authors to acknowledge the fact that the I/S-VOC to POA ratio they have used is based on very old data and have not accounted for newer estimates (e.g., May et al., 2013a,b and Jathar et al., 2014).
The Zhu et al. (2016) paper is based on modelling and it shows that good comparisons to measurements of organic concentrations are obtained with a I/S-VOC to POA emission ratio of 2.5 over Greater Paris. However, the Kim et al. (2016) paper refers to recent alkanes IVOC and SVOC tailpipe measurements, which were performed in both the gas and particle phases, for vehicles representative of the French fleet. However, it is true that although the gas/particle ratio is based on recent measurements, the volatility distribution is not, and different I/S-VOC to POA ratios could have been used for the different emission sectors. Therefore a sentence is added in section 3.1.3 for clarity. "Total I/S-VOC emissions (gas and particle phases) are estimated as detailed in Couvidat et al. (2012), by multiplying POA by a fixed value, and by assigning them to species of different volatilities. The volatility distribution is kept the same for all emission sectors, although more detailed volatility distributions could be defined following the work of May et al. (2013a, 2013b) and Jathar et al. (2014). In this study, the ratio I/S-VOC/POA is set to 2.5 (Kim et al., 2016; Zhu et al., 2016). Setting the ratio SVOC/POA to 1, i.e. ignoring I/S-VOC, has little impact on the organic concentrations, as shown in Figure 2."

4. Comment 20: This sentence needs to be rephrased since it is not clear and does not answer the posed question of how the updates change the organic aerosol mass concentrations: "During the summer 2013, the added surrogates contribute to 15% of the OM1 mass for ELVOC, 20% for organic nitrate from monoterpene oxidation and 0.2% for MBTCA."
We tried to explain that the model updates led to an increase of mass concentration, which was under-estimated when the new surrogates were simply ignored. In this version of the paper, this sentence is changed to:
"During the summer 2013, the added surrogates led to a significant increase of mass concentrations: they contributed to 15% of the OM1 mass for ELVOC, 20% for organic nitrate from monoterpene oxidation and 0.2% for MBTCA."

5. Comment 22: It's confusing when the authors talk about I/S-VOCs and their oxidation products. Are the logC* to represent the precursors or the products? If precursors, definitionally IVOCs should have C*s equal to or greater than 4. If products, how are the I/S-VOCs represented in the model? As a single model species? How is the reaction characterized? What are the reaction rates? Consider adding these details, even if brief.

The I/SVOCs are represented by 6 surrogates and using three volatility bins for the primary surrogates and three bins for the secondary surrogates. Table below shows the log(C*) of each surrogate. Because only 3 classes of volatility are taken into account for all I/S-VOC precursors, their log(C*) is taken in the low range (log(c*) = 3.5).

| Surrogates | Log(C*) at 298 K |
|---|---|
| POAlP (primary SVOC of low volatility) | -0.04 |
| POAmP (primary SVOC of medium volatility) | 1.94 |
| POAhP (primary SVOC of high volatility) | 3.51 |
| SOAlP (secondary SVOC of low volatility) | -2.04 |
| SOAmP (secondary SVOC of medium volatility) | -0.06 |
| SOAhP (secondary SVOC of high volatility) | 1.51 |

The ageing scheme of the primary surrogates is modelled using one oxidation step leading to the formation of lower volatility secondary surrogates:
POAlP + OH -> SOAlP
POAmP + OH -> SOAmP
POAhP + OH -> SOAhP

with  k = 2.0 * 10-11 cm3. molecule-1. s-1

The summary of section 2.1 is modified as follow: "As detailed in Couvidat et al. (2012), I/S-VOC emissions are emitted as three primary surrogates of different volatilities (characterized by their saturation concentrations C* : log(C* ) = -0.04, 1.93, 3.5). The ageing of each primary surrogate is represented through a single oxidation step, without NOx-dependance, to produce a secondary surrogate of lower volatility (log(C* ) = -2.4, -0.064, 1.5 respectively) but higher molecular weight. "

References:

Couvidat, F., Debry, É., Sartelet, K., and Seigneur, C.: A hydrophilic/hydrophobic organic (H2O) model: Model development, evaluation and sensitivity analysis, J. Geophys. Res., 117, D10 304, doi:10.1029/2011JD017214, 2012.

Hayes, P. L., Carlton, A. G., Baker, K. R., Ahmadov, R., Washenfelder, R. A., Alvarez, S., Rappenglück, B., Gilman, J. B., Kuster, W. C., de Gouw, J. A., Zotter, P., Prévôt, A. S. H., Szidat, S., Kleindienst, T. E., Offenberg, J. H., Ma, P. K., and Jimenez, J. L.: Modeling the formation and aging of secondary organic aerosols in Los Angeles during CalNex 2010, Atmos. Chem. Phys., 15, 5773-5801, doi:10.5194/acp-15-5773-2015, 2015.

Jathar SH, Gordon TH, Henningan C.J., Pye HOT, Pouliot G, Adams PJ, Donahue NM, Robinson AL. (2014). Unspeciated organic emissions from combustion sources and their influence on the secondary organic aerosol budget in the United States. Proc Natl Acad Sci,  111(29), 10473–10478, 2014, doi:10.1073/pnas.1323740111.

Kim, Y., Sartelet, K., Seigneur, C., Charron, A., Besombes, J.-L., Jaffrezo, J.-L., Marchand, N., and Polo, L.: Effect of measurement protocol on organic aerosol measurements of exhaust emissions from gasoline and diesel vehicles, Atmos. Environ., 140, 176–187, doi:10.1016/j.atmosenv.2016.05.045, 2016.

May, A.A. and Presto, A.A. and Hennigan, C.J. and Nguyen, N.T. and Gordon, T.D. and Robinson, A.L.. Gas-particle partitioning of primary organic aerosol emissions: (1) Gasoline vehicle exhaust. Atmospheric Environment, 77, 128-139, 2013a, doi: 10.1016/j.atmosenv.2013.04.060.

May, A.A. and Presto, A.A. and Hennigan, C.J. and Nguyen, N.T. and Gordon, T.D. and Robinson, A.L.. Gas-Particle Partitioning of Primary Organic Aerosol Emissions: (2) Diesel Vehicles. Env. Sci. & Tech, 47, 15, 8288-8296, 2013b, doi : 10.1021/es400782j.

Zhu, S., Sartelet, K., Healy, R., and Wenger, J.: Simulation of particle diversity and mixing state over Greater Paris: A model-measurement inter-comparison, Faraday Discuss., 189, 547–566, doi:10.1039/C5FD00175G, 2016.

Chrit et al. have performed simulations using an air quality model and compared predictions of organic aerosol mass and composition to measurements at a remote site in the Mediterranean Sea. They found that model updates based on the inclusion of new secondary organic aerosol formation pathways improved the model-measurement comparison. Overall the manuscript is well motivated, researched, and discussed. I recommend publication after the authors have had an opportunity to respond to my comments and considered my suggestions.

1. Page 1, line 5: Consider using the noun form: hydrophilicity
"oxidation state and hydrophilic" is replaced in the revised version by "oxidation state and hydrophilicity"

2. Page 1, line 15: 'percent' not 'percents'.
"Percents" is replaced in the revised paper by "Percent".

3. Page 2, lines 1-10: References are dated. Consider newer references.
Jimenez et al., 2009, Lin et al., 2014 and Tuet et al., 2017 are added in the revised version.

4. Page 2, lines 8-9: Rephrase.
In the revised version, we replaced the sentence "Considering health effects, oxidative stress, which is induced by the generation of reactive oxygen species (ROS), is suggested as one pathway of OA toxicity." by "In terms of health effects, OA toxicity is linked to the oxidative stress which is induced by the ROS (Reactive Oxygen Species)".

5. Page 2, lines 13: Intermediate-volatility organic compounds are a separate precursor category. Add discussion in the introduction. A re
The sentences "OA are classified either as primary (POA) or as secondary aerosols (SOA). POA are directly emitted in the atmosphere, whereas SOA are produced through chemical oxidation of volatile organic compounds (VOCs) and secondary semi-volatile organic compounds (SVOCs). SOA are often semi volatile, i.e. they partition between the gas and particle phases." are replaced by "OA are usually classified either as primary (POA) or as secondary aerosols (SOA). POA are directly emitted in the atmosphere, often as intermediate/semi-volatile organic compounds (I/S-VOCs), which partition between the gas and the particle phases (Robinson et al., 2007). The gas-phase I/S-VOC are missing from emission inventories (Couvidat et al. 2012, Kim et al. 2016). SOA are produced through chemical oxidation of volatile organic compounds (VOCs) and I/S-VOCs, and condensation of I/S-VOCs.

6. Page 2, lines 16-22: It would be helpful to be more quantitative when citing earlier work. For example, what fraction of the organic aerosol that El Haddad et al. (2011, 2013) measured was biogenic in nature?

The sentence "El Haddad et al. (2011, 2013) attributed most of the organic carbon (OC) mass to biogenic secondary organic carbon (BSOC), and to monoterpene oxidation products." is replaced by "El Haddad et al. (2011, 2013) attributed 80% of the organic aerosol mass to biogenic secondary organic aerosols (BSOA), and they attributed near 40% of the BSOA to monoterpene oxidation products."

7. Page 2, lines 17-19: Hayes et al. (2013, 2015) argue that the biogenic SOA found in Los Angeles was produced near the source and then transported into the city. Suggest citing and reconciling with Hayes work.
A reference to the work of Hayes et al. (2015) is added in the revised paper. Hayes et al. (2013, 2015) characterized the organic aerosol composition and sources in Pasadena in California during the 2010 CalNex campaign and found a substantial contribution from regional biogenic SOA. Biogenic emissions are transported and age during transport, for example during transport over the Central Valley (where there are anthropogenic emissions) in the case of Pasadena.
A sentence is added to the manuscript as followed to cite the work of Hayes et al. (2015) "A large fraction of emitted VOCs is biogenic, especially in the western Mediterranean in summer, when solar radiation is high. Biogenic emissions may age and form SOA as they are transported through different environments (Hayes et al., 2015)."

8. Page 2, line 20, Page 9, line23 (and elsewhere too): 'Fossil' and not 'fossile'.
"Fossile" is replaced by "fossil" throughout the revised paper.

9. Page 2, line 19: Was Minguillon a measurement or modeling study?
Minguillon et al. (2016) is an experimental study carried out in Barcelona in summer 2013. For clarity, the sentence "Similar results were obtained in a campaign in the Barcelona region (Spain), where Minguillón et al. (2011) have found a prevalence of non-fossil organic aerosol sources in remote and urban environments, and a clear evidence of biogenic VOC oxidation products and biogenic SOA formation under anthropogenic stressors (Minguillón et al., 2016)." is replaced by "Similar results were obtained through measurement campaigns in the Barcelona region (Spain), where Minguillón et al. (2011, 2016) have found a prevalence of non-fossil organic aerosol sources in remote and urban environments, and a clear evidence of biogenic VOC oxidation products and biogenic SOA formation under anthropogenic stressors."

10. Page 3, line 16: Incomplete sentence: 'monoterpenes oxidation products SOA over the U.S.'.
In the revised version, "organic nitrate accounts for more than a half of the monoterpene oxidation products SOA over the U.S." is replaced by "organic nitrate accounts for more than a half of the monoterpene oxidation products in the particle phase over the U.S."

11. Page 4, Section 2.1: Can you briefly summarize the existing SOA precursors, species, and processes in the model and discuss how those earlier processes do not overlap with the updates made in this work?
In section 2.1 of the revised paper, a brief description of the existing SOA anthropogenic precursors, their aging processes and products is added.
"As detailed in Couvidat et al. (2012), I/S-VOC emissions are emitted in three volatility classes (characterized by their saturation concentrations c*: log(c*) = -0.04, 1.93, 3.5). Their ageing is represented through a single oxidation step, without $NO_x$-dependence, to produce species of lower volatilities (log(c*) = -2.4, -0.064, 1.5) but higher molecular weights. For

aromatic compounds, toluene and xylene are used as surrogate precursors. The precursors react with OH to form radicals that may then react differently under low-NOx and high-NOx conditions. Under low-NOx conditions, the surrogate is not identified, but it is supposed to be hydrophobic. Under high-$NO_x$ conditions, the surrogate formed are two benzoic acids (methyl nitro benzoic acid and methyl hydroxyl benzoic acid).

12. Page 5, line 26-28: Am I correct that these oxygenated peroxy radicals can be formed only in the absence of NO? If yes, specify.

Peroxy radicals $RO_2$ are preferentially formed in low-$NO_x$ conditions, where O3 concentration is high. However, depending on the NOx concentrations, they may still form in high-NOx conditions, and then react with NO to form organic nitrate (reaction (A9) of Appendix A). Organic nitrate from ELVOCs were not considered here because their concentrations were negligible.

13. How is gas/particle partitioning of the explicit oxidation products modeled? If only briefly, please summarize the partitioning model and assumptions.

A brief description of the gas/particle partitioning of the surrogates is added in section 2.1 of the revised paper. The sentence "For organic aerosols, the partitioning is computed using SOAP (Couvidat and Sartelet, 2015), and bulk equilibrium is also assumed for SOA partitioning." is replaced by "For organic aerosols, the gas/phase partitioning of the surrogates is computed using SOAP (Couvidat and Sartelet, 2015), and bulk equilibrium is also assumed for SOA partitioning. The gas/phase partitioning of hydrophobic surrogates is modelled following Pankow (1994), with absorption by the organic phase (hydrophobic surrogates). The gas/phase partitioning of hydrophylic surrogates is computed using the Henry's law modified to extrapolate infinite dilution conditions to all conditions using an aqueous-phase partitioning coefficient, with absorption by the aqueous phase (hydrophilic organics, inorganics and water). Activity coefficients are computed with the thermodynamic model UNIFAC (UNIversal Functional group, Fredenslund et al., 1975)."

14. Page 8, line 2: I did not understand the meaning of 'nested rep.' in parentheses.
In the revised version "(nested rep.)" is replaced by "nested respectively" and "(6 June and 8 July 2012 resp.)" by "(6 June and 8 July 2012 respectively)".

15. Page 8, line 14: 'split' not 'splitted'.
"splitted" is replaced by "split" in the revised version.

16. Page 8, line 18-20: The SVOC/POA ratios might be a little too high. See work of May et al. (2014a,b) for estimates on SVOC/POA ratios. Also, is there a reason why IVOC emissions were not considered? See work of Jathar et al. (2014) and Zhao et al. (2015, 2106) for estimates of IVOCs from combustion sources as a function of VOC emissions.

IVOCs are considered in our emissions and are included as part of SVOCs. Moreover, in this paper, the I/S-VOC/POA ratio corresponds to the ratio of (gas+ particle phases)/(particle phase). This ratio equals to 1.5 if only the gas-phase of I/S-VOC is considered in the ratio I/S-VOC/POA. The ratio equals 2.5 if both gas and particle phases of I/S-VOC are considered in the ratio I/S-VOC/POA. The choice made is based on the study of Kim et al 2006 who evaluated experimentally the ratio for exhaust emissions from gasoline and diesel vehicles, and from the air-quality simulations of Zhu et al. (2016) over Greater Paris.

For clarity, the sentences "POA are assumed to be the particle phase of semi-volatile anthropogenic organic emissions (SVOC). Total SVOC emissions are estimated as detailed in Couvidat et al. (2012), by multiplying POA by a fixed value, and by assigning them to species

of different volatilities. In this study, the ratio SVOC/POA is set to 2.5 (Kim et al., 2016; Zhu et al., 2016). Setting the ratio SVOC/POA to 1 has little impact on the organic concentrations" are replaced by "POA are assumed to be the particle phase of I/S-VOC. Total I/S-VOC emissions (gas and particle phases) are estimated as detailed in Couvidat et al. (2012), by multiplying POA by a fixed value, and by assigning them to species of different volatilities. In this study, the ratio I/S-VOC/POA is set to 2.5 (Kim et al., 2016; Zhu et al., 2016). Setting the ratio I/S-VOC/POA to 1 has little impact on the organic concentrations, as shown in Figure 2."

17. Page 8, lines 25-28: What is the organic fraction in sea salt emissions?
The sentence "The organic fraction of sea-salt emissions is not taken into account in the simulation presented here. However, it is estimated in section 5, where the contribution of organic sea-salt emissions to organic concentrations is assessed." is added after line 28 of page 8 to the revised version of the paper.

18. Page 10: How does the model perform in predicting the diurnal variations in OA?
The figure below illustrates the model-to-measurement comparison of the diurnal concentrations of $OM_1$ concentrations. There is no clear diurnal variation in the measurements, probably because the organic compounds are very oxidized and of low volatility. The model slightly underestimates the concentrations of $OM_1$ especially from 00h to 10h. The slight increase in SOA concentrations in the model from 07h is due to the increase of VOC emissions and photo-oxidation during day time.

[Figure]

19. Figure 2: The composition in Figure 3 tells me that the model updates must also have increased mass concentrations and resulted in better comparison in Figure 2. This fact is missing in Section 4.1. Please state the importance of this either in Section 4.1 or in the conclusions in Section 6.
This remark is added in section 6 (conclusion) of the revised paper. "During the summer 2013, the added surrogates contribute to 15% of the $OM_1$ mass for ELVOC, 20% for organic nitrate from monoterpene oxydation and 0.2% for MBTCA. In agreement with [14]C measurements, most of the organic aerosol is from non-fossil (biogenic) origin."

20. Section 4.2: Has the model been evaluated for other pollutants? For example, black carbon, carbon monoxide, ozone, NO.

The model was also evaluated for other pollutants, like the concentrations of other particle components and precursors, and ozone. The ozone is slightly overestimated (the measured and simulated means are 51.30 μg/m³ and 60.69 μg/m³ respectively). The black carbon in PM$_{2.5}$ is also evaluated and found to be overestimated (the measured and simulated means are 0.28 μg/m³ and 0.56 μg/m³ respectively).
Another paper about the origins of particles and comparisons to measurements of the concentrations of particle components and precursors is about to be submitted. Therefore, these comparisons are not added to this paper.

21. Page 11, line 20: How low are the sesquiterpene emissions compared to isoprene and monoterpenes? Be quantitative.

In section 3.1.3 of the revised paper, the following sentence is added "… from Nature (MEGAN, Guenther et al (2006)). Over, the Mediterranean domain, during the period of the 2013 summer simulation, the mean emissions of sesquiterpenes, monoterpenes and isoprene are 0.001, 0.019 and 0.024 μg/m²/s respectively. Hence, comparing to isoprene and. monoterpene emissions, the sesquiterpene emissions are lower by a factor of 95.8% and 94.7% respectively."

22. Page 11, line 22: IVOCs are mentioned here but are they actually included since the methods do not talk about them.
As detailed in the previous replies, in the ACPD version of the paper, SVOC emissions referred to both I/S-SVOC. Details on the modelling are added: "As detailed in Couvidat et al. (2012), I/S-VOC emissions are emitted in three volatility classes (characterized by their saturation concentrations c*: log(c*) = -0.04, 1.93, 3.5). Their ageing is represented through a single oxidation step, without NO$_x$-dependence, to produce species of lower volatilities (log(c*) = -2.4, -0.064, 1.5) but higher molecular weights."

23. Figure 3 and 6, left panel: Why is the pie not a circle? Also, consider increasing the font size for readability.
The setting to include the figures in the paper is modified, so that the pie looks like a circle. The font-size in figures 3 and 6 is increased for readability sake.

24. Page 12, line 5: 'There are a variety. . .'.
"There are variety …" is modified to "There is a variety …"

25. Page 12, line 7: 'ponderating'?
"… ponderating …" is replaced in the revised paper by "… weighting …"

26. Figure 4: The legend makes it seem like the model predictions systematically discount (or subtract) the effects of the three updates rather than the opposite that is mentioned in the caption. Replace '-' with '+'?
In the legend of figure 4 of the revised paper, the "-" are replaced by "+".

27. Figure 4: Given that the OM:OC and O:C measurements are not directly measured but rather interpreted from the ACSM data, can the measurements be shown with error bars?
As stated by Crenn et al. (2015) from the intercomparison of 13 Q-ACSMs (including the one used in this study), an important instrument-to-instrument variability is observed in the O/C ratio. This variability currently remains unexplained; it appears to be independent of the

organic mass concentrations and could be due to instrument-dependent differences in the vaporization conditions. Because of this instrument-to-instrument variability, and because it is difficult to define a reference instrument which can provide an accurate measurement of the O/C ratio, a correct estimate of our Q-ACSM measurement uncertainty of the O/C ratio remains hard to provide.

However, and further to the discussion provided in the manuscript, an indicative OM/OC ratio can be calculated here from the direct comparison of OM (from Q-ACSM) and OC (from OCEC Sunset field instrument). Comparison was performed during the campaign, for a 3-week period (15/07-05/08/2013), and showed a slope of 2.0 (r²=0.80; N = 252 valid data points).

[Figure]

OC measurements were obtained at PM2.5 instead of PM1 for OM (Q-ACSM). This may lead to higher OM/OC ratio (in PM1). Based on co-located OC size-segregated measurements performed during the campaign by low-pressure 13-stage DEKATI cascade impactor, we have calculated that OC in PM2.5 was typically 10% higher compared to OC in PM1 (J. Sciare, personal communication). This would result in an experimentally determined OM/OC ratio of 2.2, which appears to be slightly above the one determined by Q-ACSM but closer to those simulated by the model.

28. Page 14, lines 5-7: Can the authors describe the mechanism at play here and the correct citation?

A brief description of the gas/particle partitioning of the surrogates is added in section 2.1 of the revised paper: "For organic aerosols, the gas/phase partitioning of the surrogates is computed using SOAP (Couvidat and Sartelet, 2015), and bulk equilibrium is also assumed for SOA partitioning. The gas/phase partitioning of hydrophobic surrogates is modelled following Pankow (1994), with absorption by the organic phase (hydrophobic surrogates). The gas/phase partitioning of hydrophylic surrogates is computed using the Henry's law modified to extrapolate infinite dilution conditions to all conditions using an aqueous-phase partitioning coefficient, with absorption by the aqueous phase (hydrophilic organics, inorganics and water). Activity coefficients are computed with the thermodynamic model UNIFAC (UNIversal Functional group, Fredenslund et al., 1975).

Therefore, the concentrations in the aqueous phase increase when the concentrations of inorganics (particularly sulfate) increase.

In section 4.3 of the revised paper, the sentence "Furthermore, a large part of biogenic SOA is hydrophilic and therefore higher condensation of sulfate enhances their partitioning into the particulate phase." is replaced by "Furthermore, a large part of biogenic SOA is hydrophilic and therefore higher condensation of sulfate enhances their partitioning into the particulate phase, as the mass of the aqueous phase increases through the condensation of sulfate (Couvidat and Sartelet (2015)."

29. Section 4.4 and Page 1, line 11: The claims about improving the hydrophilicity predictions are slightly misleading since what the authors have actually done is improve predictions of mass concentrations of water soluble organic carbon.
Yes, indeed. The improvement concerns the mass concentrations of water-soluble organic carbon. "… oxidation state and hydrophilic properties …" is replaced by "… oxidation property of organics and the hydrophilic organic carbon…" in the revised version of the paper.

30. Figure 6 does not have a left-right panel but a top-bottom panel. Fix caption.
The panels in Figure 6 are modified to not be top-bottom but left-right.

31. Figures 7, 8 and 9 could be combined into a single multi-panel figure. Also, consider adding city names to orient the reader not familiar with that part of the world.
Figures 7, 8, and 9 are combined into a multi-panel figure. Besides, the island name (Corsica), the Mediterranean sea, the Adriatic sea, the name of FRANCE and ITALY are added in order to orient the reader who are not familiar with that part of the world.

32. Sensitivity analysis: How sensitive are the model predictions and the findings from this work to the various inputs (reaction rate constants, yields, etc.) listed in the appendix? I would encourage the authors to perform additional simulations to (i) develop lower and upper bounds on their estimates for ELVOCs, organic nitrates, and MBTCA and (ii) develop insight on the most important inputs that would guide future laboratory work.

There are uncertainties in the reactions rate constants and yields of the mechanisms leading to the formation of ELVOCs, organic nitrate and MBTCA. However, those uncertainties are difficult to evaluate.

For the formation of ELVOCs and the autoxidation mechanism added in the model, the reaction rate constants are defined in Ehn et al. (2014). In fact, $k_1$ of the ozonolysis reaction is suggested by MCM, $k_{H/O2}$ is calculated in Ehn et al. (2014) to reproduce the turnover at the same point as in the observations during the chamber experiments. It is not straightforward to determine lower and upper bounds of these reaction rate constants. Concerning the yields of ELVOCs, it is possible to infer a lower and upper bound from the papers of Ehn et al. (2014) et Jokinen et al. (2015), as shown in the Table below.

| VOCs | Ehn et al. (2014) | Jokinen et al. (2015) |
|---|---|---|
| α-pinene | 7% ± 3.5% | 3.4% ± 1.7% |
| Limonene | 17% ± 8.5% | 5.3% ± 2.6% |

Details on the ELVOC yield and the choice of the bounds are added in section 2.2. The sentence "ELVOC is assumed to be formed with an average molar yield of 11% following Ehn et al. (2014), although Jokinen et al. (2015) reported lower yields (about 5%)." is replaced by "The ELVOC yield is assumed to be 11%, i.e. close to the average of the yields

of a-pinene and limonene according to Ehn et al. (2014). Jokinen et al. (2015) suggested lower yields (Table 2). In this paper, sensitivity simulations with a lower bound of 3% and a upper bound of 18% are also conducted".

The following sentences are added in section 4. "Two sensitivity simulations are performed using a lower bound yield (3%) and an upper bound yield (18%). In Appendix D, similarly to what is presented in this section for the reference simulation, the sensitivity simulations are compared to each other and to the measurements in terms of the mass of $OM_1$, the organic aerosol composition, the OM:OC and O:C ratios.

In the organic nitrate formation mechanism developed by Pye et al. (2015), the rate constants are from saprc07 (Carter et al. (2010), Hutzell et al. (2012)) except for the reaction with $HO_2$ which is calculated based on MCM for a species with 10 carbons (Jenkin et al. (1997), Saunders et al. (2003)). The yields of the reactions are taken from SAPRC07 (Carter et al. (2010)). The yields through the TERPNRO2 + NO and $NO_3$ pathways were calculated from estimates of the extent to which the radicals decompose to release $NO_2$ versus retain the nitrate group (Carter et al. (2010)). The yield from the reaction with $HO_2$ produced 100% hydroperoxide (consistent with MCM, (Jenkin et al. (1997), Saunders et al. (2003)). Therefore, it is not easy to determine lower and upper bounds of the rate constants and yields of the reactions. However, as pointed out to in Pye et al., (2015), there is a critical need for additional laboratory data when it comes to the organic nitrate sources (ozonolysis, photooxidation and chemical pathways that do not contain nitrogen). Besides, structure-dependant and pH-dependant hydrolysis rate constants and the resulting product volatility for a range of organic nitrates from biogenic VOCs oxidation.

The formation of MBTCA is very low here. Therefore, as stated in the conclusion, further sensitivity studies should not focus on determining a single yield for MBTCA, as done here, but a yield for carboxylic acids that may partition to the particle phase.

References:

Carter, W. P. L., Development of the SAPRC-07 chemical mechanism. Atmos. Environ. 2010, 44(40), 5324-5335.

Carter, W. P. L. Development of the SAPRC-07 Chemical Mechanism and Updated Ozone Reactivity Scales, Final report to the California Air Resources Board Contract No. 03−318. January 27, 2010,www.cert.ucr.edu/~carter/SAPRC.

Couvidat, F., Debry, É., Sartelet, K., and Seigneur, C.: A hydrophilic/hydrophobic organic (H2O) model: Model development, evaluation and sensitivity analysis, J. Geophys. Res., 117, D10 304, doi:10.1029/2011JD017214, 2012

F. Couvidat and K. Sartelet, The Secondary Organic Aerosol Processor (SOAP v1.0) model: a unified model with different ranges of complexity based on the molecular surrogate approach, Geosci. Model Dev., 8, 1111–1138, 2015, doi:10.5194/gmd-8-1111-2015

V. Crenn , J. Sciare , P. L. Croteau , S. Verlhac , R. Fröhlich , C. A. Belis , W. Aas , M. Äijälä, A. Alastuey, B. Artiñano, D. Baisnée, N. Bonnaire, M. Bressi, M. Canagaratna, F. Canonaco, C. Carbon, F. Cavalli, E. Coz M. J. Cubison J. K. Esser-Gietl, D. C. Green, V. Gros, L. Heikkinen, H. Herrmann, C. Lunder, M. C. Minguillón, G. Mocnik , C. D. O'Dowd,

J. Ovadnevaite, J.-E. Petit1, E. Petralia, L. Poulain, M. Priestman, V. Riffault, A. Ripoll, R. Sarda-Estève, J. G. Slowik, A. Setyan, A. Wiedensohler, U. Baltensperger, A. S. H. Prévôt, J. T. Jayne, and O. Favez: ACTRIS ACSM intercomparison – Part 1: Reproducibility of concentration and fragment results from 13 individual Quadrupole Aerosol Chemical Speciation Monitors (Q-ACSM) and consistency with co-located instruments, Atmos. Meas. Tech., 8, 5063–5087, doi:10.5194/amt-8-5063-2015, 2015.

Ehn, M., Thornton, J., Kleist, E., Sipilä, M., Junninen, H., Pullinen, I., Springer, M., Rubach, F., Tillmann, R., Lee, B., Lopez-Hilfiker, F., Andres, S., Acir, I., Rissanen, M., Jokinen, T., Schobesberger, S., Kangasluoma, J., Kontkanen, J., Nieminen, T., Kurtén, T., Nielsen, 10 L. B., Jørgensen, S., Kjaergaard, H. G., Canagaratna, M., Dal Maso, M., Berndt, T., Petäjä, T., Wahner, A., Kerminen, V., Kulmala, M., Worsnop, D. R., Wildt, J., and Mentel, T. F.: A large source of low-volatility secondary organic aerosol, Nature, 506, 476–479, doi:10.1038/nature13032, 2014.

Fredenslund, Aa., Jones, R. L. and Prausnitz, J. M., Group-Contribution Estimation of Activity Coefficients in Nonideal Liquid Mixtures. AIChE J., 21, 1086-1099 (1975).

Jenkin, M. E.; Saunders, S. M.; Pilling, M. J., The tropospheric degradation of volatile organic compounds: A protocol for mechanism development. Atmos. Environ. 1997, 31 (1), 81-104.

Jokinen, T., Berndt, T., Makkonen, R., Kerminen, V., Junninen, H., Paasonen, P., Stratmann, F., Herrmann, H., Guenther, A. B., Worsnop, D. R., Kulmala, M., Ehn, M., and Sipiläb, M.: Production of extremely low volatile organic compounds from biogenic emissions: Measured yields and atmospheric implications, Proc. Nat. Acad. Sci., 112, 7123–7128, doi:10.1073/pnas.1423977112, 2015.

Hayes, P. L., Carlton, A. G., Baker, K. R., Ahmadov, R., Washenfelder, R. A., Alvarez, S., Rappenglück, B., Gilman, J. B., Kuster, W. C., de Gouw, J. A., Zotter, P., Prévôt, A. S. H., Szidat, S., Kleindienst, T. E., Offenberg, J. H., Ma, P. K., and Jimenez, J. L.: Modeling the formation and aging of secondary organic aerosols in Los Angeles during CalNex 2010, Atmos. Chem. Phys., 15, 5773-5801, doi:10.5194/acp-15-5773-2015, 2015.

Hutzell, W. T.; Luecken, D. J.; Appel, K. W.; Carter, W. P. L., Interpreting predictions from the SAPRC07 mechanism based on regional and continental simulations. Atmos. Environ. 2012, 46 (0), 417- 429.

Nieminen, T., Kurtén, T., Nielsen, 10 L. B., Jørgensen, S., Kjaergaard, H. G., Canagaratna, M., Dal Maso, M., Berndt, T., Petäjä, T., Wahner, A., Kerminen, V., Kulmala, M., Worsnop, D. R., Wildt, J., and Mentel, T. F.: A large source of low-volatility secondary organic aerosol, Nature, 506, 476–479, doi:10.1038/nature13032, 2014.

Pye, H., Luecken, D. J., Xu, L., Boyd, C., Ng, N., Baker, K., Ayres, B., Bash, J., Baumann, K., Carter, W., Edgerton, E., Fry, J., Hutzell, W., 10 Schwede, D., and Shepson, P.: Modelling the current and the future roles of particulate organic nitrates in the southeastern United States, Environ. Sci. Technol., 49, 14 195–14 203, doi:10.1021/acs.est.5b03738, 2015.

Saunders, S. M.; Jenkin, M. E.; Derwent, R. G.; Pilling, M. J., Protocol for the development of  the Master Chemical Mechanism, MCM v3 (Part A): tropospheric degradation of non-aromatic volatile organic compounds. Atmos. Chem. Phys. 2003, 3, 161-180.